# ACTIONS-TO-ACTION: INDUCTIVE ATTENTION FOR EGOCENTRIC VIDEO ACTION ANTICIPATION

## ABSTRACT

Video action anticipation is a specific field within computer vision that diverges from action recognition, requiring the prediction of future actions through the analysis of historical video sequences. This paper unveils an innovative model designed to overcome the limitations of existing solutions by amalgamating recurrent and attention mechanisms, taking cues from the principles of object tracking. Notably, our model leverages prior anticipation results, enabling a nuanced interpretation of semantic transitions between actions and recognizing the uncertainty inherent in predicting future events. This strategy strikes a balance between computational efficiency and judicious data utilization, challenging the assumptions prevalent in current transformer models and thereby underlining its practicality for real-world applications. Distinctively, our model discerns temporal connection from abstract concepts in a way that mirrors human reasoning and adopts a recurrent structure to thoroughly capture video context. Extensive experiments conducted on EPIC-Kitchens-100, EPIC-Kitchens-55, and EGTEA Gaze+ confirm the superior performance of our proposed model and efficiency compared to established transformer architectures. Remarkably, our proposed model surpasses most multi-modal models by only using RGB visual inputs, showcasing its exceptional generalization capabilities across a variety of unseen test sets.

## 1 INTRODUCTION

Accurately anticipating upcoming events is integral to human decision-making and routine planning. Yet, endowing machines with this innate ability remains a tough challenge, marking a crucial advancement in the field of video understanding. The challenge is intensified by the imperative to unravel uncertainties about future events through meticulous examination of historical data, particularly in the nuanced domain of video action anticipation. This endeavor significantly deviates from action recognition, necessitating the forecasting of impending actions by scrutinizing historical sequences within video data. Furthermore, video action anticipation frequently employs egocentric videos, which harmonize perspectives from diverse subjects and implicitly unveil their intentions. This is achieved by integrating elements such as coarse-grained visual attention, indicated by the camera's heading direction, within the observed frames. These prerequisites are prevalent across various practical applications of video-based prediction, including assistive navigation systems OhnBar et al. (2018), collaborative robotics Park et al. (2016), interactive entertainment Liang et al. (2015); Taylor et al. (2020), and autonomous vehicles Hirakawa et al. (2018).

While action recognition focuses on classifying current actions through pattern recognition, video action anticipation uses these patterns to predict the complex nature of potential future actions, each with multiple possibilities. This complexity is further exacerbated by the essential many-to-many mapping between past and future actions, a departure from the conventional many-to-one mapping in action recognition. This distinction underscores the intricate nature of video action anticipation, paving the way for groundbreaking research in this rapidly evolving field.

Many models initially developed for action recognition have been repurposed, but using identical architectures for both domains has yielded suboptimal results. Recent research largely focuses on utilizing transformer models to tackle the inherent challenges of video action anticipation Girdhar & Grauman (2021); Wu et al. (2022). However, this approach faces two main limitations: a fixed receptive field for past data hinders continuous inference in real-world scenarios, and the inherent as-

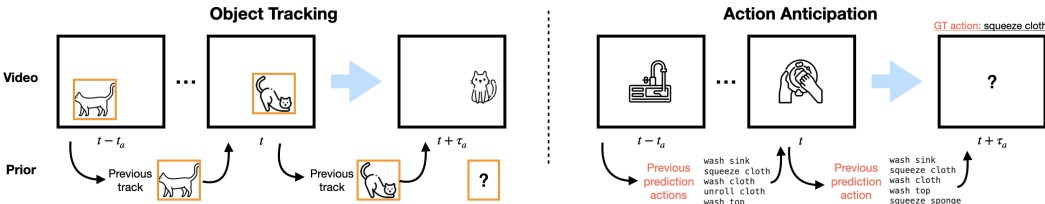

Figure 1: Similar to how object tracking leverages the properties of previous track results as a prior for subsequent predictions, we propose a method in action anticipation that also utilizes previous anticipation results, aiming to better capture the semantic movement conveyed by past history.

sumption within the attention mechanism does not consistently align with video action anticipation, where inputs are merely tentative indicators of future events.

Inspired by the strategies employed in object tracking Carion et al. (2020); Zhu et al. (2020), where subsequent predictions are informed by prior ones, we integrate previous anticipation results as priors to refine future action forecasting (see Figure 1). This integration enables our model to recognize semantic transitions between actions and addresses the limitation of using only the latest visual input as the query. Our model detects video patterns to predict future actions, using past conditions to enhance understanding of complex interactions, focusing more on overall patterns than on specific details like pixels. Nevertheless, given the temporal dependencies inherent to actions, video action anticipation necessitates more than a single frame to capture the inherent changes present in tracking applications. Therefore, we incorporate a recurrent structure to assimilate video context, enabling the model to adaptively use a dynamic receptive field and perform continuous online inferencing.

In summary, our contributions are threefold:

- We introduce a novel model for video action anticipation that melds recurrent and attention mechanisms, demonstrating superior accuracy across various benchmarks.
- Informed by insights from object tracking, our model explicitly employs prior anticipation results to refine subsequent action predictions, thereby significantly enhancing model generalizability.
- Experimental results and in-depth analyses indicate that our model achieves competitive performance gains and suggests a promising direction for future models in the field. Additionally, it maintains an effective balance in model size and computational efficiency.

## 2 RELATED WORK

Our model draws inspiration from recurrent neural networks (RNNs) and self-attention mechanisms to forecast future actions. In this section, we briefly review the literature on modeling with RNN and Transformers in the context of the action anticipation problem.

**RNNs in Action Anticipation.** Recurrent Neural Networks (RNNs) have been extensively applied in modeling sequential inputs, with numerous studies affirming their effectiveness in action prediction tasks Wang et al. (2018); Su et al. (2020). Early work in video action anticipation incorporated Long Short-Term Memory (LSTM) networks within an encoder-decoder framework, establishing foundational benchmarks Furnari & Farinella (2020a). Subsequent developments include augmenting the original LSTM encoder to account for spatial-temporal structures Osman et al. (2021), introducing self-regulated modules for leveraging long-range context features Qi et al. (2021), and exploring label smoothing to address future uncertainties Camporese et al. (2021). Variants of the recurrent mechanism have also been investigated, with works predicting forthcoming frames and integrating future information Wu et al. (2020); Fernando & Herath (2021), employing higher-order recurrence with spatial-temporal decomposed attention Tai et al. (2022a), and offering a generalized perspective of the recurrent mechanism as message-passing learning Tai et al. (2022b).

**Transformers in Action Anticipation.** Transformers, with self-attention at their core, have achieved significant advancements in video action recognition Bertasius et al. (2021); Arnab et al. (2021); Vaswani et al. (2017). Originally designed for language processing, Vision Transformers (ViTs) have become prominent across various visual tasks Dosovitskiy et al. (2020); Zhou et al.

(2021); Han et al. (2022). In the realm of action anticipation, transformer-based models have demonstrated their capability to outperform recurrent baselines. Notable examples include models integrating ViT backbones and causal decoders to encapsulate temporal dependencies Girdhar & Grauman (2021), and models incorporating memory compression modules for efficient long-range dependency modeling Wu et al. (2022). Transformers exhibit versatility in modeling sequential inputs Jaegle et al. (2021b;a). In line with this, our model integrates attention as a dynamic mechanism for extracting temporal context.

## 3 PROBLEM STATEMENT

Video action anticipation is defined as a task where an arbitrary length of video inputs $X = \{x_i \in \mathbb{R}^{C \times H \times W} | i = T - t_s, \ldots, T\}$, starting from timestep $T - t_s$ to $T$, is used to predict the target action that happens in the $\tau_a$ seconds. To support the continuous inference, at every timestep $t$, the model uses the observed input up to time $t$, denoted as $X_{*:t}$, to predict the subsequent action, $y_{t+\tau_a} \in \mathbb{R}^{N_c}$, where $N_c$ is the number of classes, denotes the future action occurring after $\tau_a$ seconds. For the benchmark dataset, we measure the performance at the latest timestep $T$ and compared with ground-truth future action $y_{T+\tau_a}$.

Existing methods, whether recurrent or transformer-based, model to maximize the predict probability $P(y_{T+\tau_a} | X_{*:T}, \{\psi, \theta\})$, parameterized by learnable weights $\psi$ and $\theta$, through the estimated action probability of the last timestep $\hat{y}_T$ and compare with annotated ground-truth action probability $y_{T+\tau_a}$, as shown below:

$$P(y_{T+\tau_a} | X, \{\psi, \theta\}) = \underbrace{P(y_{T+\tau_a} | \hat{y}_T)}_{\text{anticipation loss}} \underbrace{P(\hat{y}_T | h_T, \psi)}_{\text{classifier}} \underbrace{P(h_T | X_{*:T}, \theta)}_{\text{spatio-temporal model}} . \tag{1}$$

For recurrent-based modeling, $P(h_T | X_{*:T}, \theta)$ is presented by a recurrent update $h_t = f_\theta(x_t + g(h_{t-1}))$ where $t = T - t_s, \ldots, T$ and $g(.)$ is an arbitrary gate function which fits. On the other hand, for the transformer-based modeling, $P(h_T | X_{*:T}, \theta)$ is built by multi-layer self-attentions.

To align the prediction with the ground-truth annotation of a future action, a cross-entropy loss is applied between the prediction $\hat{y}_t$ and the corresponding future action probability $y_{t+\tau_a}$ as given by:

$$\mathcal{L}_{CE}(y_{t+\tau_a}, \hat{y}_t) = -\Sigma_{t=T-t_s}^{T} w_t(y_{t+\tau_a} \log \hat{y}_t). \tag{2}$$

The loss function aims to minimize empirical risk across the entire sequence. Here, $w_t$ represents the loss weight per timestep, set to 2 at $t = T$ and 1 otherwise, except for unannotated timesteps $y_{i+\tau_a}$ where $w_t$ is 0, often due to non-action movements or sparsely annotated datasets.

## 4 OUR APPROACH

The core of our design focuses on enhancing the hidden state representation by integrating prior predictions. In the context of equation 1, the hidden state $h_T$ is utilized for modeling the video dependency based on low-level video frames, a method which may be sub-optimal. To address this, we introduce a modification whereby the hidden state is also explicitly conditioned on prior action predictions $\widehat{y_{*:T-1}}$ as inputs. This modification is depicted in the following equation:

$$P(y_{T+\tau_a} | X, \{\psi, \theta\}) = P(y_{T+\tau_a} | \hat{y}_T) P(\hat{y}_T | h_T, \psi) P(h_T | X_{*:T}, \widehat{y_{*:T-1}}, \theta) \tag{3}$$

Our model innovatively merges attention with recurrent structures, enhancing effectiveness through higher-order recurrent updates and targeted attention on pivotal hidden states, based on their action domain relevance. This fusion, we termed **inductive attention**, concisely encapsulates historical action trends to forecast future actions.

### 4.1 HIGHER-ORDER RECURRENT STRUCTURE

To effectively utilize the historical information in the video sequence and address the issue of forgetfulness in recurrent networks, we augment the recurrent network from a first-order to a higher-order structure.

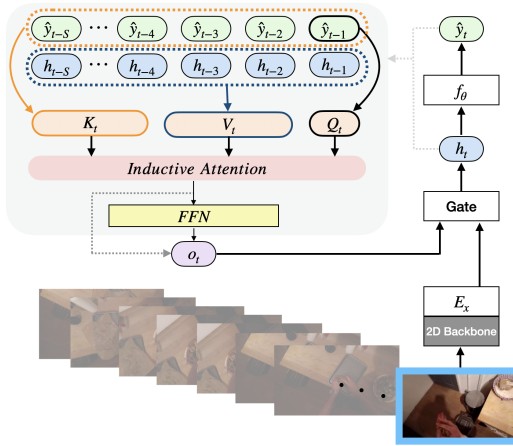

**Algorithm 1** Inductive Attention Model

**Given:** input $X$; compression functions $E_x, E_q, E_k$; classifier $f_\psi$.
**Initialize:** $M_0 \leftarrow \varnothing$ with maximum capacity $S$.
**for** every time $t$ receiving input $x_t \in X$ **do**
    $e_t \leftarrow E_x(x_t)$
    **if** $t = 0$ **then**
        $o_t \leftarrow 0$
    **else**
        Read $\widehat{y_{t-S:t-1}}$ and $h_{t-S:t-1}$ from $M_{t-S:t-1}$
        $Q_t \leftarrow E_q(\widehat{y_{t-1}})$
        $K_t \leftarrow E_k(\widehat{y_{t-S:t-1}})$
        $V_t \leftarrow h_{t-S:t-1}$
        $o_t \leftarrow Ind\text{-}Attn(Q_t, K_t, V_t)$
    **end if**
    $g_t \leftarrow \sigma\left(w_2^T\left(\max(0, [o_t; e_t]w_1)\right)\right)$
    $h_t \leftarrow g_t \cdot o_t + (1 - g_t) \cdot e_t$
    $\hat{y}_t \leftarrow f_\psi(h_t)$
    $M_t \leftarrow$ push $E_K(\text{sg}(\hat{y}_t))$ and $h_t$ into $M_{t-1}$
    **if** $M_t$ exceeds the maximum capacity $S$ **then**
        Pop the oldest element in $M_t$
    **end if**
    **yield** $\hat{y}_t$
**end for**

Figure 2: The Inductive Attention Model adopts the prior prediction, $\widehat{y_{t-1}}$, as the query and calculates its correlation with the attention keys $(\widehat{y_{t-1}}, \ldots, \widehat{y_{t-S}})$, aiming to aggregate the higher-order recurrent states $(h_{t-1}, \ldots, h_{t-S})$. Subsequently, the output merges with the encoded frame input through a gating function, resulting in the formation of the recurrent state $h_t$, which is utilized to compute the current prediction $\widehat{y_t}$.

Analogous to traditional n-gram language models Shannon (1948), higher-order recurrent networks generate new outputs and internal states by consolidating multiple preceding states, thereby extending beyond the first-order Markov chain assumption inherent in conventional recurrent networks.

To elaborate, in a first-order recurrent model, the hidden state $h_t$ is calculated as:

$$h_t = f(x_t + g(h_{t-1})). \tag{4}$$

Conversely, in a higher-order recurrent model, several past states are referenced and amalgamated to determine the new state:

$$h_t = f(x_t + \Phi(h_{t-1}, \ldots, h_{t-S})). \tag{5}$$

Here, $S$ denotes a hyperparameter determining the number of past states (the order of the model).

Various alternatives are available for the aggregation function $\Phi$, including linear function Soltani & Jiang (2016), polynomial function Yu et al. (2017), convolutional tensor-train decomposition Su et al. (2020), and spatial-temporal decomposed attention Tai et al. (2022a).

In this study, we have adapted the attention mechanism to function as $\Phi$. Unlike the conventional attention definition Vaswani et al. (2017), our approach integrates previous action predictions $\widehat{y_{*:T-1}}$ as the query (Q) and also for the keys (K) of attention for guiding the focus with explicit exposing of action semantics.

## 4.2 INDUCTIVE ATTENTION

The standard form of attention Vaswani et al. (2017) computes the dot-product correlation of keys $K$ conditioned on the query $Q$ to retrieve the correlated context in the values $V$:

$$Attn(Q, K, V) = softmax\left(\frac{Q^T K}{\sqrt{d}}\right) V, \tag{6}$$

where $d$ denotes the dimension of vectors $K$ and $Q$.

In equation 6, $Q$ represents the current interests used to retrieve the correlated topics in the values $V$, while the key $K$ provides the locations where information can be retrieved from $V$ by interacting with $Q$. A core modification in our inductive attention is to encode $K$ with the corresponding results of the predicted actions $\widehat{y_{*:T-1}}$. Additionally, we assign $\widehat{y_{T-1}}$ to $Q$ instead of using $x_{T-1}$, explicitly injecting action semantics into equation 6.

For implementation, at every timestep $t$, given the trajectory of previous higher-order recurrent states, $h_{t-1}, h_{t-2}, \ldots, h_{t-S}$, we first compress the prediction $\widehat{y_{t-1}}$ as query $Q$ using a learnable compression function $E_q$. Correspondingly, we compress $\widehat{y_{t-S:t-1}}$ into key $K$ via $E_k$, and assign the recurrent states as the value $V$. Notably, the inductive attention is used to summarize the higher-order information (e.g., $\Phi(.)$ in equation 5), so the lengths of $K$ and $V$ are limited to a finite length $S$ and maintained by a queue with a first-in-first-out (FIFO) update policy.

In summary, the inductive attention can be expressed as follows:

$$Ind\text{-}Attn(\hat{y}_{t-1}, Mt) \coloneqq Attn(Q_t, K_t, V_t), \tag{7}$$

$$where\ Q_t = E_q(\widehat{y_{t-1}}), \tag{8}$$

$$K_t = E_k(\widehat{y_{t-S:t-1}}), \tag{9}$$

$$V_t = h_{t-1:t-S}. \tag{10}$$

For simplicity, we omit the embeddings for $Q_t$, $K_t$, and $V_t$.

### 4.3 INDUCTIVE ATTENTION MODEL

At each timestep $t$, the frame feature $e_t$ is initially encoded by $E_x$. Typically, a backbone is utilized within the encoder to extract these frame features. The recurrent state is then formulated in accordance with higher-order equation 5, incorporating $e_t$ as the input and utilizing inductive attention as the aggregation function:

$$e_t = E_x(x_t), \tag{11}$$

$$o_t = Ind\text{-}Attn(\underbrace{Q_t, K_t, V_t}_{\text{as per eqs } (8)(9)(10)}), \tag{12}$$

$$h_t = g_t e_t + (1 - g_t) o_t, \tag{13}$$

$$\widehat{y}_t = f_\psi(h_t). \tag{14}$$

Here, $g_t = \sigma\left(w_2^T \max(0, w_1[o_t; e_t])\right)$ is a gate function governs the balance between past experiences (derived from the output of inductive attention) and the input from the current frame. The sigmoid function $\sigma(\cdot)$ maps values to the interval $[0, 1]$. Notably, we assign $o_t = 0$ for the initial step $t = 0$. The forwarding process is succinctly summarized in Algorithm 1 and Figure 2.

Significantly, our inductive attention model executes a many-past-to-many-future estimation while calculating the dot product $Q^T K$ in equation 7, employing $Q \equiv \widehat{y_{t-1}}$ and $K \equiv \widehat{y_{t-1:t-S}}$.

## 5 EXPERIMENTS

### 5.1 IMPLEMENTATION DETAILS

For all experiments, we employed TSN Furnari & Farinella (2020b), ConvNeXt Liu et al. (2022), and Swin Liu et al. (2021) as the backbone variants in our model. We trained our model using the AdamW optimizer Loshchilov & Hutter (2017), with a learning rate of 2e-4 and a cosine decay scheduler. The weight decay was set to 1e-2, excluding biases and normalization layers. The model was trained with a batch size of 128 for 50 epochs on a single NVIDIA RTX 3090 GPU, with automatic mixed precision enabled. Our model incorporates a single IAM layer with a hidden size $d = 2048$ for equation 13, and a dropout rate of 0.6. The remaining hyperparameters are detailed individually for each dataset.

### 5.2 DATASETS

EPIC-Kitchens-100 (EK100) Damen et al. (2022) is a substantial dataset featuring 100 hours of egocentric videos, 3806 action labels, 67217 training segments, and 9668 validation segments. We adopted a class weighting approach, akin to Wu et al. (2022). The Mean Top-5 Recall (MT5R) for action, verb, and noun is assessed at an anticipation interval $\tau_a = 1s$, with each sequence sampled at 1 fps to compile 30 frames.

EPIC-Kitchens-55 (EK55) Damen et al. (2018) includes 55 hours of recordings, 2513 action classes, 23492 training segments, and 4979 validation segments. We applied label smoothing Müller et al.

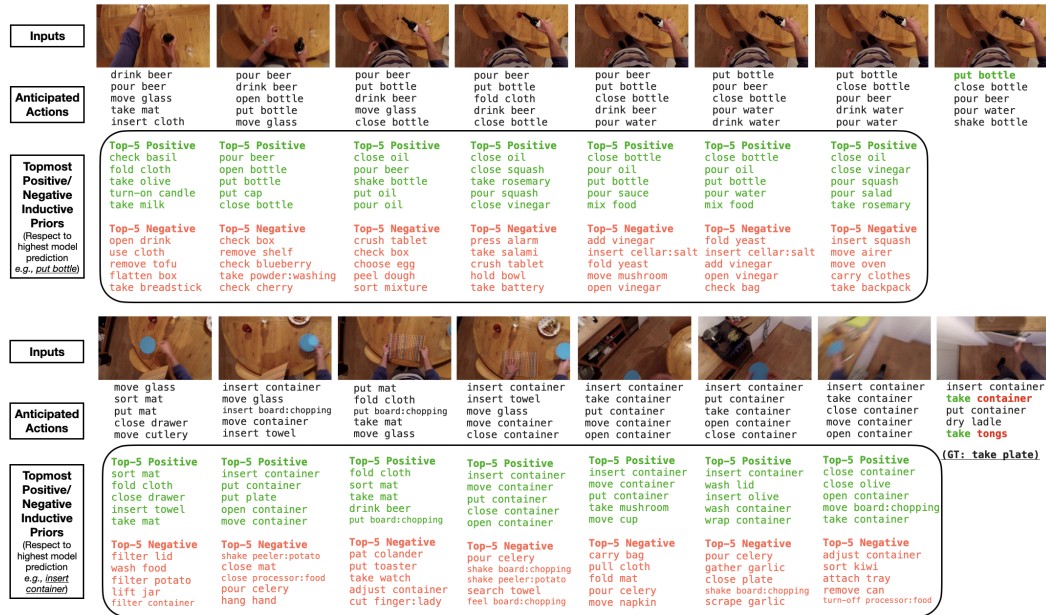

Figure 3: Illustrating two examples from the EK100 dataset, each with eight consecutive frames. Top-5 action anticipations and corresponding inductive priors are displayed below each frame.

(2019) with a ratio of 0.6 and reported Top-1/5 action accuracy and MT5R at $\tau_a = 1s$. The dataset was sampled at 1 fps, resulting in 10 frames per sequence.

EGTEA Gaze+ Li et al. (2018) consists of over 28 hours of recordings, annotated with 106 unique action classes across 10321 segments. Following the protocol in Girdhar & Grauman (2021), we evaluated on split 1, comprising 8299 training clips and 2022 validation clips. The anticipation accuracy is gauged at $\tau_a = 0.5s$ for top-1 action accuracy and mean top-1 action recall, with the dataset sampled at 2 fps to generate 10 frames per sequence.

## 5.3 RESULTS

### 5.3.1 EK100 ACTION ANTICIPATION

We assessed our proposed model, IAM, on the EK100 dataset against previous works, with the results consolidated in Table 1. Utilizing the TSN backbone, IAM demonstrated notable enhancement over several well-established baselines, including the former top-ranking competition winner, AVT, and displayed competitiveness with MeMViT. When the TSN was replaced with the more recent ConvNeXt (convolutional-based) and Swin (transformer-based) backbones, IAM excelled, surpassing the optimal configuration of MeMViT by +0.4 in overall classes, +1.4 in unseen classes, and +2.1 in tail classes.

Table 2 reveals that our approach outperforms state-of-the-art methods in both single-modal and some multi-modal frameworks in terms of test scores, highlighting the significant generalizability of our proposed method. The test evaluation is conducted on a held-out set, and the results are submitted to the official challenge server for assessment.

Figure 3 presents qualitative examples for analysis. The first row illustrates a scenario characterized by an extended duration and consistent action, where the content of the frames remains constant across the sequence. There is a discernible shift in predicted actions between the 5th and 6th frames, transitioning from the verb "pour" to "put." This transition underscores the importance of considering prior prediction in our model. In contrast, the second row highlights a scenario where the model incorrectly predicts the noun "plate" as "container" in the ground-truth anticipation "take plate." This example underscores the utility of egocentric videos in revealing intentions like "take" through changes in viewpoint. Beneath each action prediction, we display the top positive and negative actions identified by inductive attention as relevant to the highest prediction. Further analysis of inductive attention and additional qualitative examples are available in the supplementary material.

Table 1: EK100 validation results, measured in MT5R at $\tau_a = 1s$, detailing individual accuracy for actions (A), verbs (V), and nouns (N).

| | Methods | Param Counts | Overall Classes | | | Unseen Classes | | | Tail Classes | | |
|---|---|---|---|---|---|---|---|---|---|---|---|
| | | | A | V | N | A | V | N | A | V | N |
| Multi-Modals | TempAgg (Sener et al. (2020)) | - | 14.7 | 23.2 | 31.4 | 14.5 | 28.0 | 26.2 | 11.8 | 14.5 | 22.5 |
| | RULSTM Furnari & Farinella (2020b) | - | 14.0 | 27.8 | 30.8 | 14.2 | 28.8 | 27.2 | 11.1 | 19.8 | 22.0 |
| | TSN-AVT+ Girdhar & Grauman (2021) | - | 14.8 | 25.5 | 31.8 | 11.5 | 25.5 | 23.6 | 12.6 | 18.5 | 25.8 |
| | AVT+ Girdhar & Grauman (2021) | - | 15.9 | 28.2 | 32.0 | 11.9 | 29.5 | 23.9 | 14.1 | 21.1 | 25.8 |
| | TSN-AFFT+ Zhong et al. (2023) | - | 17.0 | 22.3 | 31.5 | 14.0 | 23.8 | 25.3 | 15.0 | 14.6 | 23.6 |
| | Swin-AFFT+ Zhong et al. (2023) | - | 18.5 | 22.8 | 34.6 | 15.5 | 24.8 | 26.4 | 16.2 | 15.0 | 27.7 |
| Single-Modal | chance | - | 0.2 | 6.4 | 2.0 | 0.5 | 14.4 | 2.9 | 0.1 | 1.6 | 0.2 |
| | TempAgg Sener et al. (2020) | - | 13.0 | 24.2 | 29.8 | 12.2 | 27.0 | 23.0 | 10.4 | 16.2 | 22.9 |
| | RULSTM Damen et al. (2022) | - | 13.3 | 27.5 | 29.0 | - | - | - | - | - | - |
| | HORST Tai et al. (2022a) | - | 13.2 | 24.5 | 30.0 | - | - | - | - | - | - |
| | MPNNEL-TB Tai et al. (2022b) | - | 14.8 | 28.7 | 31.4 | - | - | - | - | - | - |
| | TSN-AVT Girdhar & Grauman (2021) | - | 13.6 | 27.2 | 30.7 | - | - | - | - | - | - |
| | irCSN152-AVT Girdhar & Grauman (2021) | - | 12.8 | 25.5 | 28.1 | - | - | - | - | - | - |
| | AVT Girdhar & Grauman (2021) | 378M | 14.9 | 30.2 | 31.7 | - | - | - | - | - | - |
| | TSN-DCR-LSTM Xu et al. (2022) | 14M | 14.5 | 27.9 | 28.0 | - | - | - | - | - | - |
| | TSM-DCR-LSTM Xu et al. (2022) | 20M | 15.2 | 28.4 | 28.5 | - | - | - | - | - | - |
| | TSN-DCR Xu et al. (2022) | 78M | 14.6 | 31.0 | 31.1 | - | - | - | - | - | - |
| | TSM-DCR Xu et al. (2022) | 84M | 16.1 | 32.6 | 32.7 | - | - | - | - | - | - |
| | TSN-AFFT Zhong et al. (2023) | - | 16.4 | 21.3 | 32.7 | 13.6 | 24.1 | 25.5 | 14.3 | 13.2 | 25.8 |
| | Swin-AFFT Zhong et al. (2023) | - | 17.6 | 23.4 | 33.7 | 15.2 | 24.5 | 25.4 | 15.3 | 15.6 | 26.5 |
| | MeMViT 16x4 Wu et al. (2022) | 59M | 15.1 | **32.8** | 33.2 | 9.8 | 27.5 | 21.7 | 13.2 | 26.3 | 27.4 |
| | MeMViT 32x3 Wu et al. (2022) | 212M | 17.7 | 32.2 | 37.0 | 15.2 | 28.6 | 27.4 | 15.5 | 25.3 | 31.0 |
| | **TSN-IAM (Ours)** | 42M | 17.5 | 32.2 | 35.7 | 11.9 | 31.4 | 24.9 | 16.8 | **26.9** | 31.8 |
| | **ConvNeXt-IAM (Ours)** | 142M | 17.6 | 31.4 | 36.2 | 12.0 | 33.0 | 25.0 | 17.1 | 26.0 | 32.0 |
| | **Swin-IAM (Ours)** | 141M | **18.1** | 32.1 | **37.2** | **16.6** | **34.6** | **27.9** | **17.6** | 26.7 | **33.0** |

Table 2: EK100 test results, measured in MT5R at $\tau_a = 1s$. The test evaluation is conducted on a held-out set, and the results are submitted to the official challenge server for assessment.

| | Methods | Overall Classes | | | Unseen Classes | | | Tail Classes | | |
|---|---|---|---|---|---|---|---|---|---|---|
| | | A | V | N | A | V | N | A | V | N |
| Single-Modal | RULSTM Furnari & Farinella (2020b) | 11.2 | 25.3 | 26.7 | 9.7 | 19.4 | 26.9 | 7.9 | 17.6 | 16.0 |
| | TBN Zatsarynna et al. (2021) | 11.0 | 21.5 | 26.8 | 12.2 | 20.8 | 28.3 | 7.2 | 13.2 | 15.4 |
| | TempAgg Sener et al. (2020) | 12.6 | 21.8 | 30.6 | 10.5 | 17.9 | 27.0 | 8.9 | 13.6 | 20.6 |
| | AVT+ Girdhar & Grauman (2021) | 12.6 | 25.6 | 28.8 | 8.8 | 20.9 | 22.3 | 10.1 | 19.0 | 22.0 |
| | TCN-TSN Zatsarynna et al. (2021) | 10.9 | 20.4 | 26.6 | 11.1 | 17.9 | 26.9 | 7.0 | 11.7 | 15.2 |
| | TCN-TBN Zatsarynna et al. (2021) | 11.0 | 21.5 | 26.8 | 12.2 | 20.8 | 28.3 | 7.2 | 13.2 | 15.4 |
| | AFFT-TSN+ Zhong et al. (2023) | 13.4 | 19.4 | 28.3 | 9.9 | 14.0 | 24.2 | 10.9 | 12.0 | 19.5 |
| | AFFT-Swin+ Zhong et al. (2023) | 14.9 | 20.7 | 31.8 | 12.1 | 16.2 | 27.7 | 11.8 | 13.4 | 23.8 |
| | RAFTformer-2B Girase et al. (2023) | 15.4 | 30.1 | 34.1 | - | - | - | - | - | - |
| | **Swin-IAM (Ours)** | **16.4** | **30.7** | **35.1** | **13.5** | **23.0** | **29.2** | **14.3** | **25.6** | **29.9** |
| Ensemble | TransAction Gu et al. (2021) | 13.4 | - | - | 10.1 | - | - | 11.9 | - | - |
| | Panasonic (Yamamuro *et al.*) | 14.8 | 30.4 | 33.5 | 10.2 | 21.1 | 27.1 | 12.7 | 24.6 | 27.5 |
| | AVT++ Girdhar & Grauman (2021) | 16.7 | 26.7 | 32.3 | 12.9 | 21.0 | 27.6 | 13.8 | 19.3 | 24.0 |
| | DCR Xu et al. (2022) | 17.3 | - | - | 14.1 | - | - | 14.3 | - | - |

#### 5.3.2 EK55 Action Anticipation

Table 3 provides a summarized comparison of the performance between our model and previous works on the EK55 benchmark. The results demonstrate that our model is competitive in terms of top-5 action accuracy and mean top-5 recall. Similar to the EK100 evaluation, our model, when integrated with the Swin backbone, achieves the best overall performance. It is worth noting that AVT-h, when based on the ViT-based AVT-b backbone, does not demonstrate better accuracy on EK55 as it does on EK100 compared to the convolutional backbone irCSN152. This suggests that AVT does not yield consistent scores with a specific backbone across different datasets. DCR achieves higher top-1 accuracy, a potential benefit arising from their implementation of curriculum learning from future frames. It is important to mention that EK55 has sparser annotations in the sampling sequence, resulting in less information available to be leveraged in the prior predictions $\widehat{y_{t-1:t-S}}$ utilized in our model.

#### 5.3.3 EGTEA Gaze+ Action Anticipation:

We further assessed our model on the EGTEA Gaze+ egocentric video dataset, and the findings are presented in Table 4. Our best configuration, employing the Swin backbone, elevates the top-1 action accuracy and top-1 action recall, surpassing the state-of-the-art models, AVT and AFFT.

Table 3: EK55 validation results at the $\tau_a = 1s$. The top-1/top-5 action accuracy and top-5 action recall are summarized.

| Methods | Backbone | External Data | Top-1 | Top-5 | Recall |
|---|---|---|---|---|---|
| RULSTM Furnari & Farinella (2020a) | TSN | IN1K | 13.1 | 30.8 | 12.5 |
| TempAgg Sener et al. (2020) | TSN | IN1K | 12.3 | 28.5 | 13.1 |
| ImagineRNN Wu et al. (2020) | TSN | IN1K | 13.7 | 31.6 | - |
| SRL Qi et al. (2021) | TSN | IN1K | - | 31.7 | 13.2 |
| HORST Tai et al. (2022a) | TSN | IN1K | 12.8 | 31.6 | 12.2 |
| MPNNEL-TB Tai et al. (2022b) | TSN | IN1K | 13.8 | 32.0 | 13.6 |
| AVT-h Girdhar & Grauman (2021) | TSN | IN1K | 13.1 | 28.1 | 13.5 |
| AVT-h Girdhar & Grauman (2021) | AVT-b | IN21+1K | 12.5 | 30.1 | 13.6 |
| AVT-h Girdhar & Grauman (2021) | irCSN152 | IG65M | 14.4 | 31.7 | 13.2 |
| DCR Xu et al. (2022) | TSN | IN1K | 13.6 | 30.8 | - |
| DCR Xu et al. (2022) | irCSN152 | IG65M | 15.1 | **34.0** | - |
| DCR Xu et al. (2022) | TSM | IN1K | **16.1** | 33.1 | - |
| **IAM (Ours)** | TSN | IN1K | 13.5 | 32.1 | 14.3 |
| **IAM (Ours)** | ConvNeXt | IN22K | 13.3 | 32.6 | 15.1 |
| **IAM (Ours)** | Swin | IN22K | 14.2 | **34.0** | **16.1** |

Table 4: EGTEA Gaze+ validation results on the split 1 at the $\tau_a = 0.5s$. We reported the top-1 accuracy and mean top-1 recall of each individual action (A), verb (V) and noun (N).

| Methods | Top-1 Acc | | | Mean Top-1 Recall | | |
|---|---|---|---|---|---|---|
| | A | V | N | A | V | N |
| I3D-Res50 Carreira & Zisserman (2017) | 34.8 | 48.0 | 42.1 | 23.2 | 31.3 | 30.0 |
| FHOI Liu et al. (2020) | 36.6 | 49.0 | 45.5 | 32.5 | 32.7 | 25.3 |
| TSN-AVT-h Girdhar & Grauman (2021) | 39.8 | 51.7 | 50.3 | 28.3 | 41.2 | 41.4 |
| TSN-AFFT Zhong et al. (2023) | 42.5 | 53.4 | 50.4 | 35.2 | 42.4 | 44.5 |
| AVT Girdhar & Grauman (2021) | 43.0 | 54.9 | 52.2 | 35.2 | **49.9** | 48.3 |
| **TSN-IAM (Ours)** | 43.5 | 54.3 | 52.2 | 35.5 | 43.8 | 46.6 |
| **ConvNeXt-IAM (Ours)** | 44.6 | 54.5 | 53.1 | 36.3 | 42.6 | 45.3 |
| **Swin-IAM (Ours)** | **45.4** | **55.9** | **54.3** | **37.4** | 46.5 | **49.3** |

It is crucial to note that EGTEA Gaze+ only annotations for the latest timestep $T$. Consequently, $\widehat{y_{t-1:t-S}}$ in our model solely relies on self-prediction without of any supervision.

Moreover, the results indicate that models utilizing sequential or recurrent networks tend to have an edge over those based on clip approaches. These observations align with similar findings reported in Wang et al. (2018); Su et al. (2020).

## 5.4 DISCUSSION

This section delves into the analysis of the optimal context length for IAM and discusses the design choices made. All experiments related to this analysis were conducted using the EK100 dataset.

**Context Length.** We evaluated the performance across varying context lengths, ranging from 10 to 60 seconds, where longer lengths correspond to the observation of an increased number of prior action trajectories. Table 5 unveils that our model reaches its peak performance with 30-second inputs. This suggests that performance does not invariably enhance with extended contexts, a conclusion that is in harmony with preliminary studies Furnari & Farinella (2020a). However, the optimal input length for our model markedly exceeds that reported for earlier recurrent models, underscoring our model's proficiency in harnessing historical data in extended video observations through higher-order inductive attention.

Furthermore, in the EK100 dataset, a 30-second context encompasses approximately 0 to 22 actions, averaging $7.1 \pm 3.5$ actions across the training set. Figure 5 illustrates that to address the uncertainties of the future, a greater number of actions should be observed in the past trajectory. Consequently, averaging 7.1 actions can substantially mitigate the complexity of the many-past-to-many-future problem to almost a many-past-to-one-future scenario.

**Model Efficiency and Robustness.** By employing a recurrent structure to consolidate past observations and incorporating higher-order structures coupled with inductive attention to hone pertinent states, our proposed model attains paramount accuracy in action anticipation while preserving efficiency in model size. Figure 4 draws a comparison between action scores and parameter counts.

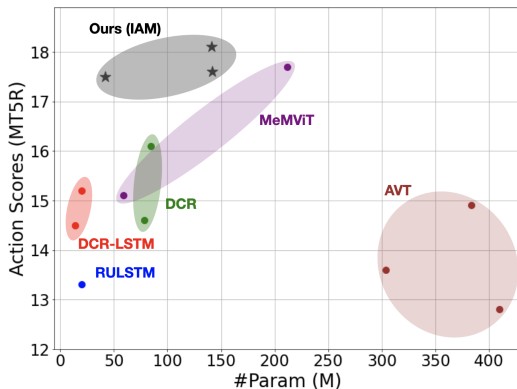

Figure 4: Comparison of IAM to previous methods on the EK100 dataset, illustrating higher action anticipation accuracy with fewer parameters. Multiple points denote variants with different backbones.

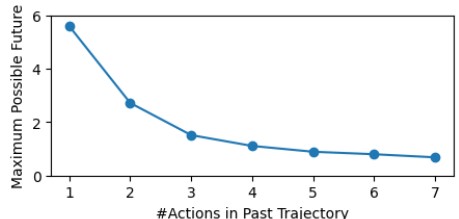

Figure 5: Quantity of potential futures in relation to the number of actions in past trajectory.

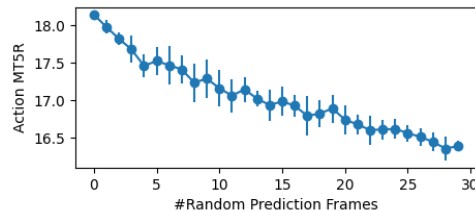

Figure 6: Evaluation of robustness through randomizing the number of prior predictions.

Table 5: Comparison of different context lengths under various backbone options.

| Context | MT5R (%) | | |
|---|---|---|---|
| Length | TSN | ConvNeXt | Swin |
| 10s | 16.9 | 16.9 | 17.2 |
| 20s | 17.5 | 17.4 | 18.0 |
| **30s** | **17.5** | **17.6** | **18.1** |
| 40s | 17.3 | 16.9 | 18.0 |
| 50s | 16.7 | 17.5 | 17.9 |
| 60s | 16.7 | 17.4 | 18.0 |

Table 6: Illustrating the performance differences by sequentially removing individual components.

| Proposed Change | MT5R (%) | Δ MT5R (%) |
|---|---|---|
| **Swin-IAM** | **18.1/16.6/17.6** | -/-/- |
| - Inductive Attention | 16.8/15.8/17.0 | -1.3/-0.8/-0.6 |
| - Class Weighting | 15.9/15.0/14.2 | -0.9/-0.8/-2.8 |
| **ConvNeXt-IAM** | **17.6/12.0/17.1** | -/-/- |
| - Inductive Attention | 16.7/11.2/16.2 | -0.9/-0.8/-0.9 |
| - Class Weighting | 15.5/12.1/13.6 | -1.2/+0.9/-2.6 |
| **TSN-IAM** | **17.5/11.9/16.8** | -/-/- |
| - Inductive Attention | 16.4/10.9/16.0 | -1.1/-1.0/-0.8 |
| - Class Weighting | 15.3/11.2/13.8 | -1.1/+0.3/-2.2 |

Moreover, Figure 6 showcases the elegant degradation in performance of the proposed model when subjected to injected random frame predictions. The values plotted for $n = 1, \ldots, 30$ are ascertained by randomizing $t - n$ prior action predictions $\widehat{y_{t-1:t-S}}$, which are utilized as the keys in inductive attention (as illustrated in equation 9), thereby exemplifying the robustness of the model to variations in the accuracy of preceding predictions.

**Model Performance.** In Table 6 analyzes the effectiveness of our innovative inductive attention mechanism across different backbone architectures. This mechanism integrates higher-order structure using predictions as priors, consistently shows improvements (refer to supplementary material for more details). Specifically, it achieves increases of +1.3, +0.9, and +1.1 in overall MT5R for Swin, ConvNeXt, and TSN backbones, respectively. Additionally, class weighting, which tackles the sensitivity of MT5R to class imbalances, contributes to further improvements. It is worth to note that although the performance gains from inductive attention may seem modest, their cumulative impact in combination with calls weighting is substantial and addresses diverse challenges within our model.

## 6 CONCLUSION

This paper introduces an Inductive Attention Model (IAM) designed for video action anticipation. By utilizing a higher-order recurrent structure, IAM efficiently captures temporal information and infers from historical data. Incorporating previous predictions as query and keys in inductive attention allows the model to handle longer contexts in videos, setting new benchmarks in accuracy on extensive egocentric video datasets. Notably, our approach surpasses existing methods by offering enhanced accuracy and efficiency, all while necessitating fewer model parameters.

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
