# Actions-to-Action: Inductive Attention for Egocentric Video Action Anticipation – Supplementary material

## A  Input Preprocessing

For the TSN baseline utilized in the three action anticipation benchmarks shown in Tables 1, 3, and 4 of the main paper, we sourced the pre-extracted features from the official EPIC-Kitchens websites[1]. TSN employs the BN-Inception backbone, extracting RGB frame input of size (456, 256). The implementation and pretrained weights of Swin and ConvNeXt, used to present results in Tables 1, 2, 3, and 4, are adopted from the open-source implementation[2]. The frame inputs for Swin and ConvNeXt are of size (224, 224), with values rescaled to the range [-1, 1].

## B  Inductive Attention Analysis

Table 7 provides an analysis of the components comprising our novel inductive attention mechanism when integrated with Swin, ConvNeXt, and TSN backbones on the EPIC-Kitchens-100 datasets. This inductive attention is characterized by two key enhancements: (a) the utilization of prior predictions as the attention query $Q$ and keys $K$, and (b) the expansion of the attention context window from first-order to higher-order historical states. Our results demonstrate that this advanced inductive attention consistently elevates performance across various backbones, as evidenced by improvements in overall, unseen, and tail action Mean Top-5 Recall (MT5R) metrics.

Table 7: Analysis of models with and without the implementation of inductive attention, focusing on their performance in overall, unseen, and tail Mean Top-5 Recall (MT5R) metrics within the EPIC-Kitchens-100 dataset.

| Proposed Change | MT5R (%) | $\Delta$ MT5R (%) |
|---|---|---|
| **With Inductive Attention (Swin-IAM)** | **18.1/16.6/17.6** | -/-/- |
| - (a) Prior predictions serve as attention query (Q) and keys (K) | 17.2/15.9/17.4 | -0.9/-0.7/-0.2 |
| - (b) Extend to higher-order (S=1 to S=30) | 16.8/15.8/17.0 | -0.4/-0.1/-0.4 |
| **Without Inductive Attention** | **16.8/15.8/17.0** | **-1.3/-0.8/-0.6** |
| **Inductive Attention on (ConvNeXt-IAM)** | **17.6/12.0/17.1** | -/-/- |
| - (a) Prior predictions serve as attention query (Q) and keys (K) | 16.9/11.9/16.4 | -0.8/-0.1/-0.7 |
| - (b) Extend to higher-order (S=1 to S=30) | 16.7/11.2/16.2 | -0.2/-0.7/-0.2 |
| **Without Inductive Attention** | **16.7/11.2/16.2** | **-1.0/-0.8/-0.9** |
| **Inductive Attention on (TSN-IAM)** | **17.5/11.9/16.8** | -/-/- |
| - (a) Prior predictions serve as attention query (Q) and keys (K) | 17.0/11.1/16.4 | -0.5/-0.8/-0.4 |
| - (b) Extend to higher-order (S=1 to S=30) | 16.4/10.9/16.0 | -0.6/-0.2/-0.4 |
| **Without Inductive Attention** | **16.4/10.9/16.0** | **-1.1/-1.0/-0.8** |

We expand our analysis to identify the most significant positive and negative inductive priors in relation to the predictions of each model. This identification is accomplished through the calculation of partial gradients in relation to the attention keys. These keys are conditioned by previous predictions, which in turn maximize the prediction being assigned. Such methodology effectively reveals the sensitivity of the priors. It is crucial to highlight that this depth of analysis is a distinctive

---

[1]https://github.com/epic-kitchens/C3-Action-Anticipation
[2]https://github.com/huggingface/pytorch-image-models, v0.5.4

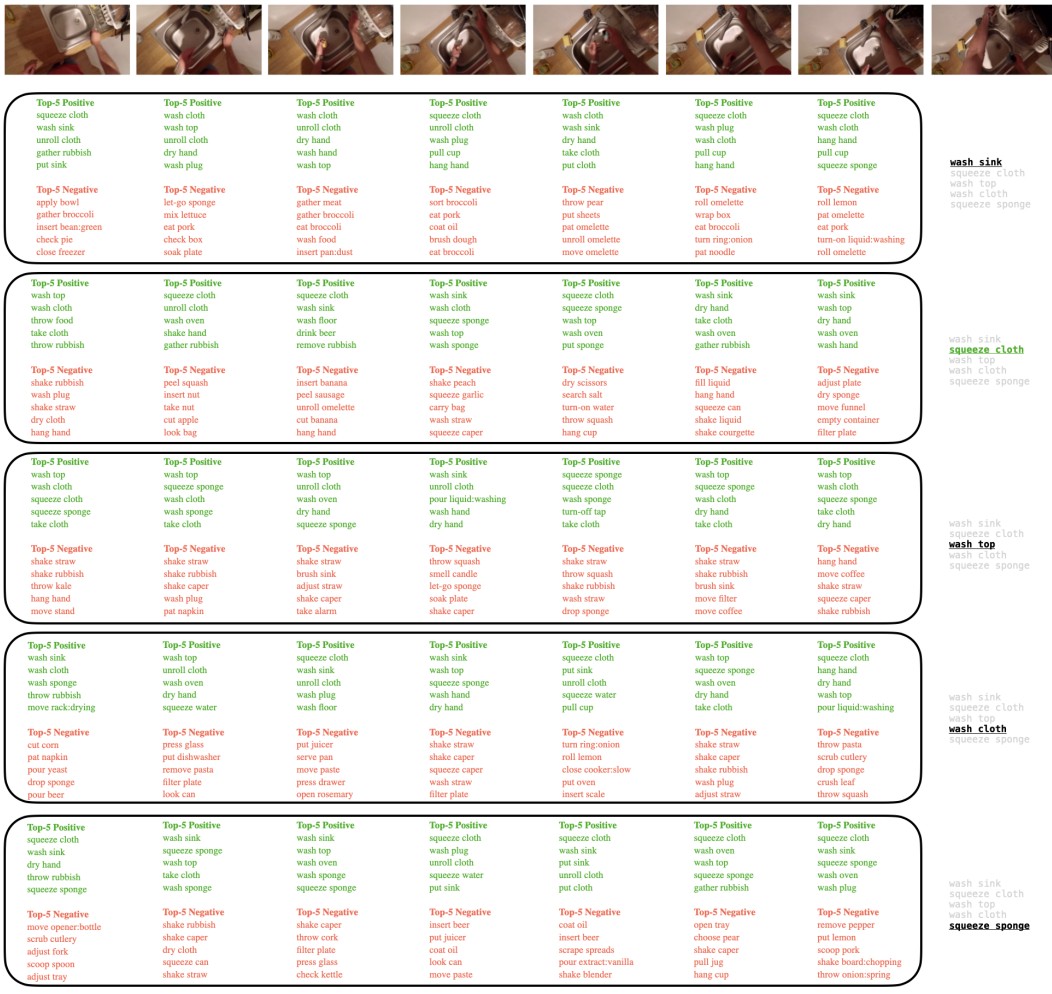

Figure 7: Illustration of a sample with correct model prediction. The top row displays the final eight frames of the observed video sequence. Directly beneath the last frame, top-5 prediction of inductive attention model is presented, with the ground truth highlighted in green. Subsequently, each of the following rows showcases the topmost positive and negative priors, computed from our inductive attention, corresponding to each of the top-5 predictions.

feature of the inductive attention mechanism we propose. This uniqueness stems from our method of explicitly representing both the query and the keys within the action space context. Thereby, this approach not only enhances the interpretability and precision of the model but also contributes to making the decision-making process of the model more transparent. This analysis is based on the Swin-IAM model on EPIC-Kitchens-100 samples.

Figure 7 presents an example of the top prior predictions contributing to each model prediction. The correct answer ranks second among the top-5 predicted actions. In this sample, a washing action is identified, followed by a squeezing action. This sequence is further corroborated by the priors contributing most to the predictions. Conversely, the topmost negative priors appear noticeably irrelevant to the described action.

An intriguing observation is the sensitive prior action "wash sink", which correlates with the targeted prediction "squeeze cloth". This correlation indicates a potential causal relationship, where "wash sink" precedes "squeeze cloth". While this pattern does not universally apply in our model analysis, it uncovers a promising avenue for future research, exploring the potential causal relationships in model predictions.

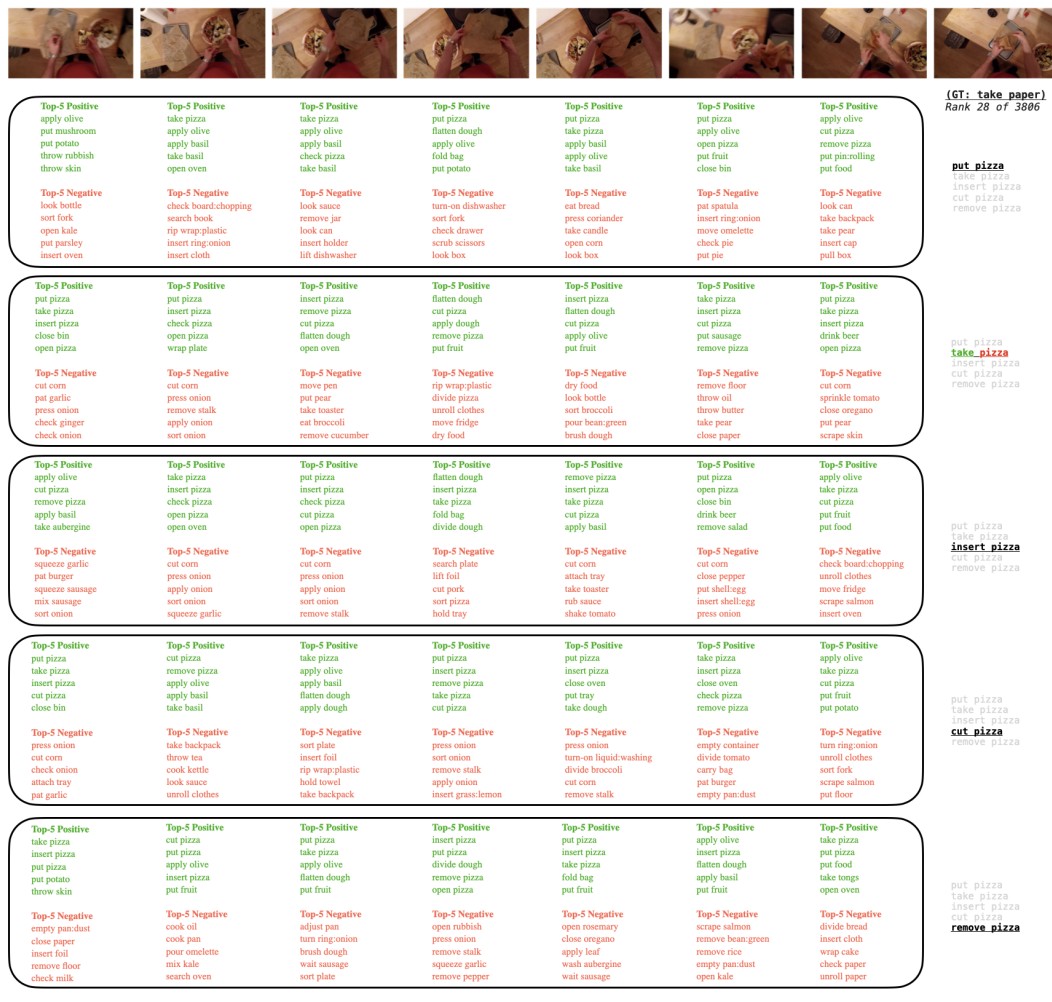

Figure 8: Illustration of a sample with incorrect model prediction. The top row displays the final eight frames of the observed video sequence. Directly beneath the last frame, top-5 prediction of inductive attention model is presented, with the ground truth highlighted in green. Subsequently, each of the following rows showcases the topmost positive and negative priors, computed from our inductive attention, corresponding to each of the top-5 predictions.

In another example, illustrated in Figure 8, the predictions of the model are notably erroneous, with the correct prediction ranking only 29th out of 3806. A closer examination of inductive prior of our model reveals a significant bias towards the noun "pizza", as seen in the video. From a human perspective, accurately interpreting this video hinges on recognizing the action of transferring a pizza from paper to a plate. This highlights a key challenge in action anticipation: discerning the underlying intention amidst distracting elements. It is noteworthy that the actual ground truth of this sample is unrelated to the pizza, but instead concerns the paper. Despite the incorrect prediction in this instance, the model output remains plausible and aligns with the inductive attention's learned priors for actions.

The per-frame top-5 action anticipation for these two cases is presented in Figure 9 (a) for the correct example and Figure 10 (a) for the incorrect one.

## C  QUALITATIVE ANALYSIS

We derive success and failure cases of our model from the EPIC-Kitchens-100, EPIC-Kitchens-55, and EGTEA Gaze+ datasets for qualitative analysis. Four video clips, labeled **(a)** to **(d)**, illustrate

instances where the ground truth is within the top-5 predictions of model at the last frame. Additionally, another four examples, labeled **(e)** to **(h)**, depict cases where the model failed to accurately anticipate the target action. For each video sample, we present the last eight frames along with the corresponding top-5 predictions per frame. The figures are most effectively viewed horizontally and with zoom-in.

## C.1 EPIC-KITCHENS-100

Figures 9 and 10 display samples from the EK100 dataset. The model demonstrates proficiency in refining predictions to specific verbs of activities based on context. For instance, in video **(a)**, predictions converge to verbs *"wash"* or *"squeeze"*. Moreover, video **(b)** initially recognizes *"milk"* and subsequently shifts to the plausible resultant object *"cereal"*. Video **(c)** contains minimal movements, with the model maintaining relevant predictions but facing challenges in anticipating the object until the final moment. Video **(d)** represents a challenging scenario with ambiguous subject intention throughout the observation.

Figure 10 uncovers instances where our model encountered difficulties. In video **(e)**, despite accurately identifying the verb, an incorrect noun *"pizza"* is consistently predicted. Other cases reflect mispredictions due to inadequate visual observations (e.g., **(f)**) or an abundance of potential objects (e.g., **(g)**, **(h)**).

## C.2 EPIC-KITCHENS-55

Figure 11 showcases successful cases, noting that the target noun objects in videos **(a)** and **(b)** are not visible. Nonetheless, the model accurately infers based on past predictions propagated by inductive attention. Additionally, video **(c)** demonstrates the model's capacity to identify the object *"plate"* appearing only in the initial three frames. Video **(d)** maintains predictions pertinent to cooking pasta.

Conversely, sample **(e)** lacks any indication of the anticipated object *"tofu"*, predicting *"container"* instead. Mispredicted nouns and verbs are also observed in videos **(f)**, **(g)**, and **(h)**.

## C.3 EGTEA GAZE+

Figure 13 highlights video **(b)**, demonstrating correct prediction through awareness of *tomato:container* visible only in the first two frames. Video **(c)** involves subtle movements about the *bread:container*, depicted as being taken out and replaced in the final three frames.

Figure 14 reveals incorrect predictions, primarily attributed to insufficient observable evidence (e.g., **(e)** and **(f)**) or an array of possibilities (e.g., **(g)**, with notable confidence on verbs), or a combination as in **(h)**.

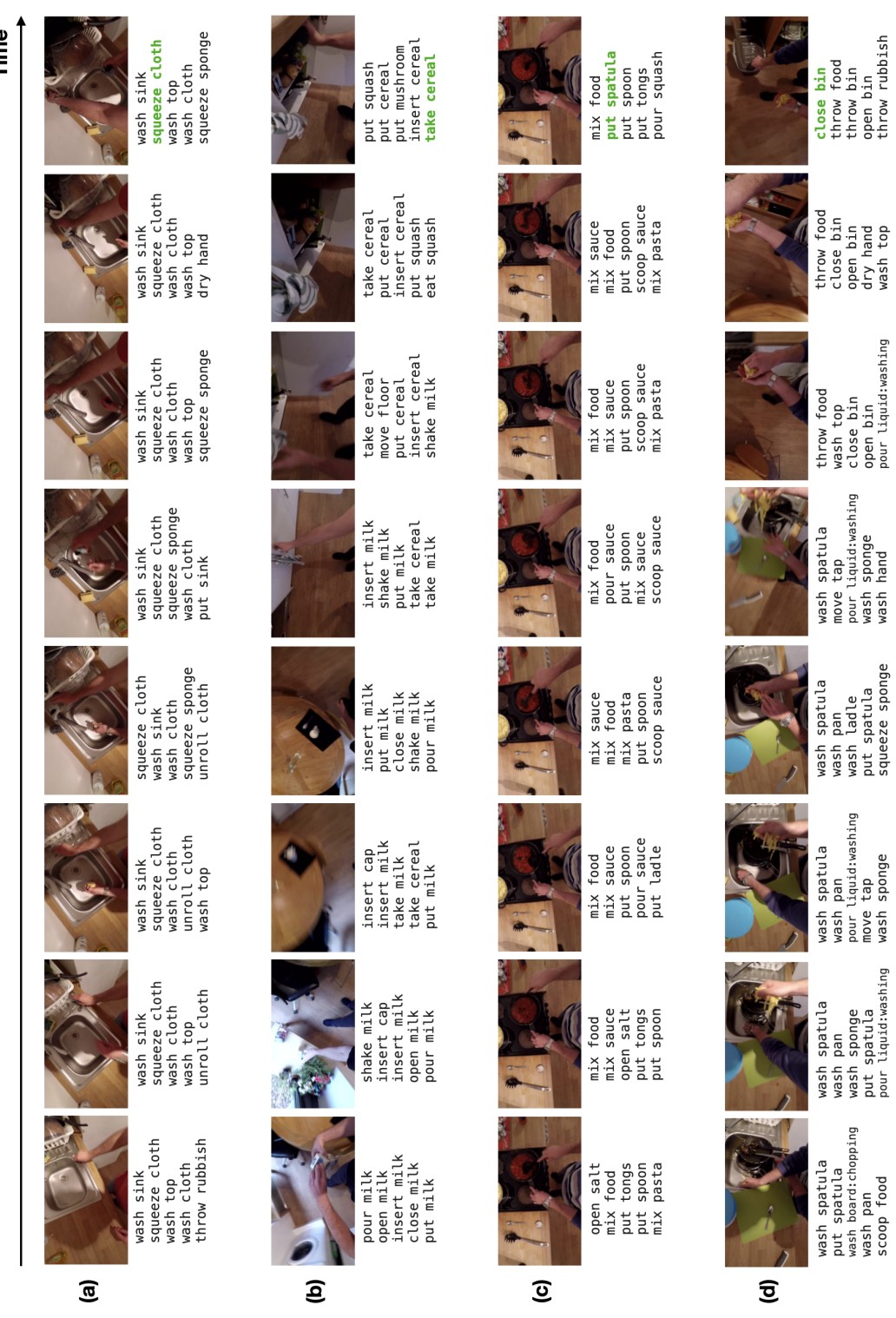

Figure 9: Four video clips from the **EPIC-Kitchens-100** validation set, illustrating **correct** predictions, are presented in the figure. Each clip displays the last eight frames along with the corresponding top-5 action anticipations at $\tau_a = 1s$. The ground truth is prominently highlighted in bold green.

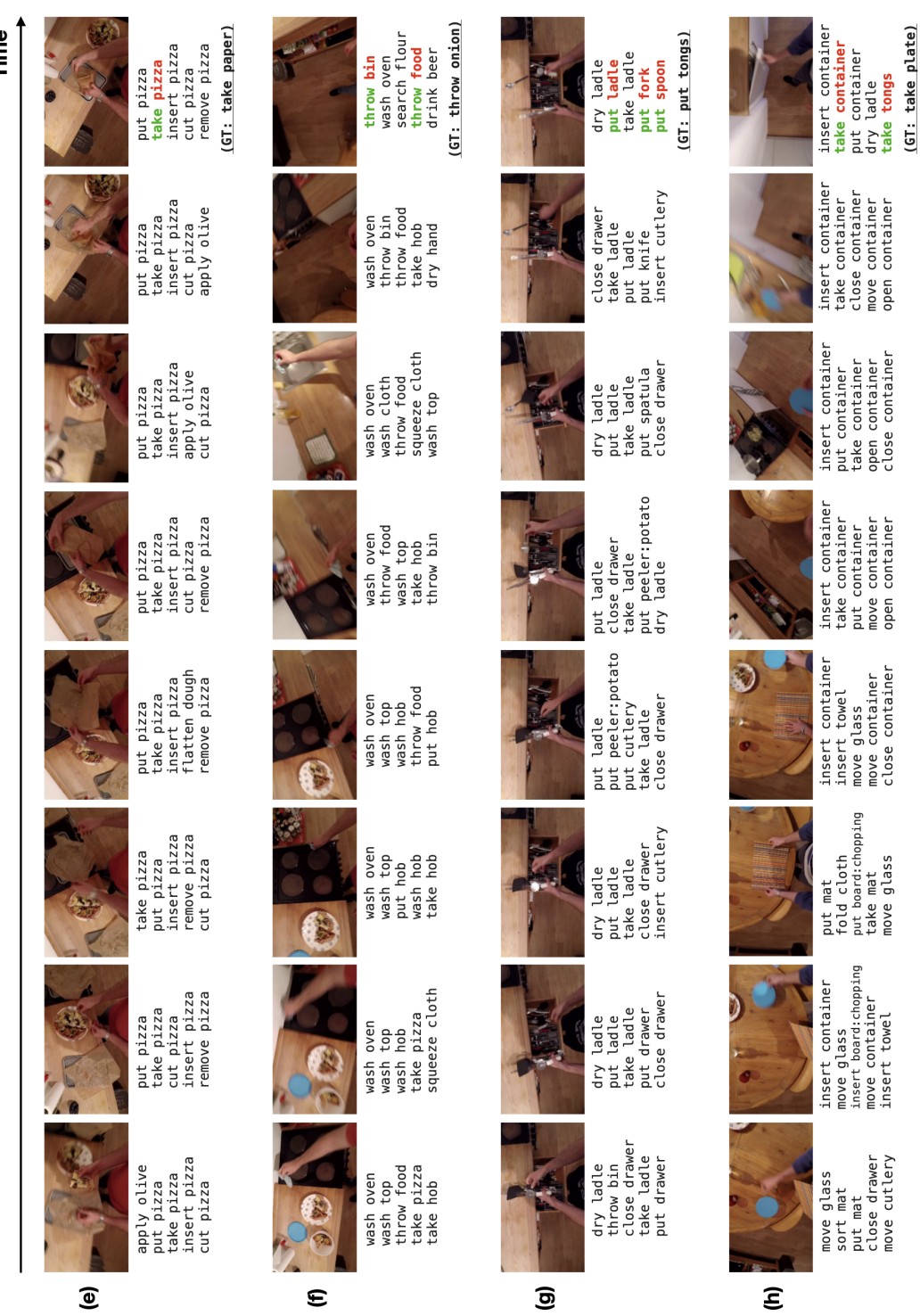

Figure 10: The figure displays four video clips from the **EPIC-Kitchens-100** validation set, showcasing **incorrect** predictions. Each clip features the last eight frames and the corresponding top-5 action anticipations at $\tau_a = 1s$. The ground truth is revealed in the final frame, with the correct verb and noun of the action highlighted in bold green.

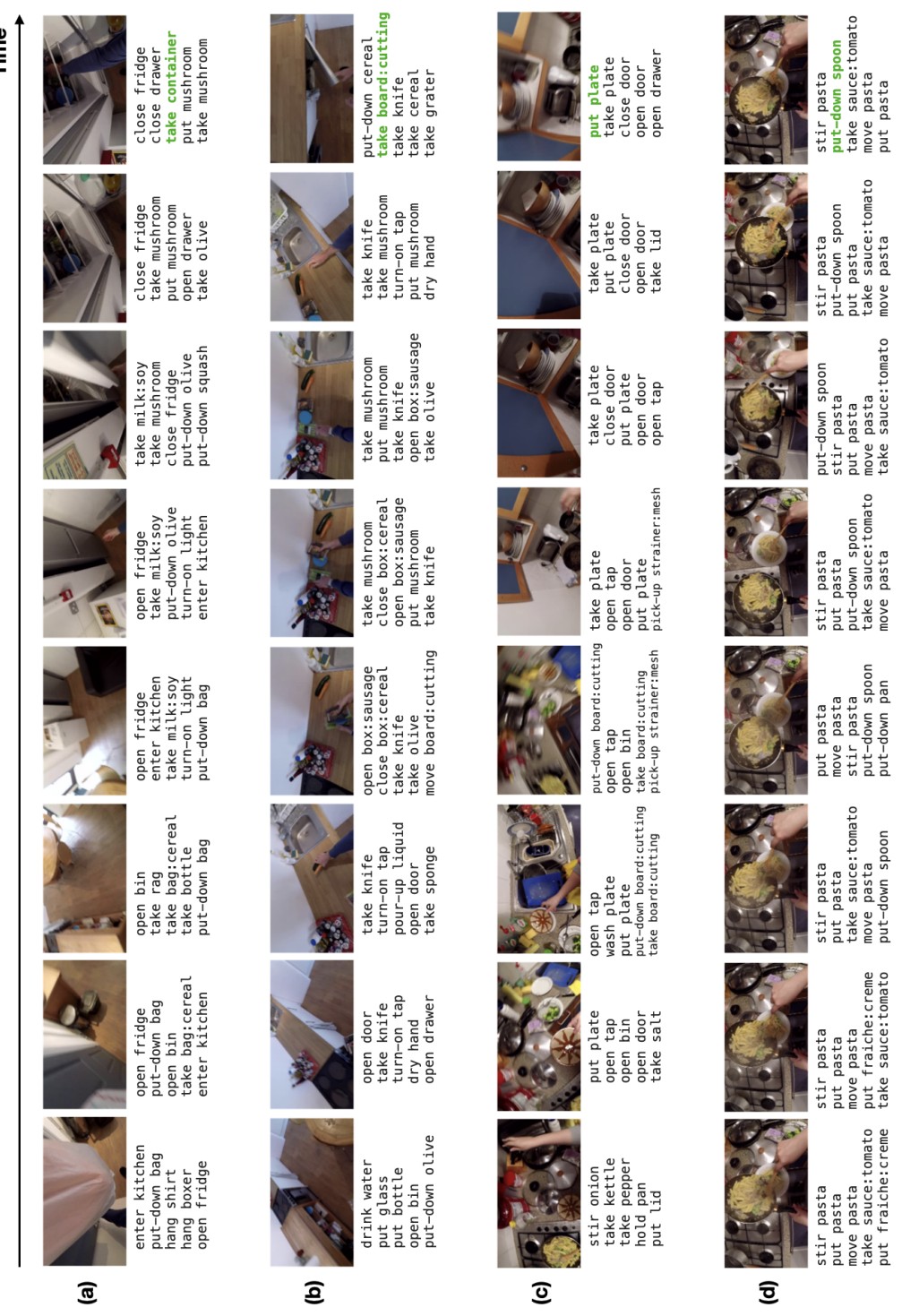

Figure 11: The figure displays four video clips from the **EPIC-Kitchens-55** validation set, illustrating **correct** predictions. Each clip features the last eight frames along with the corresponding top-5 action anticipations at $\tau_a = 1s$. The ground truth is prominently highlighted in bold green.

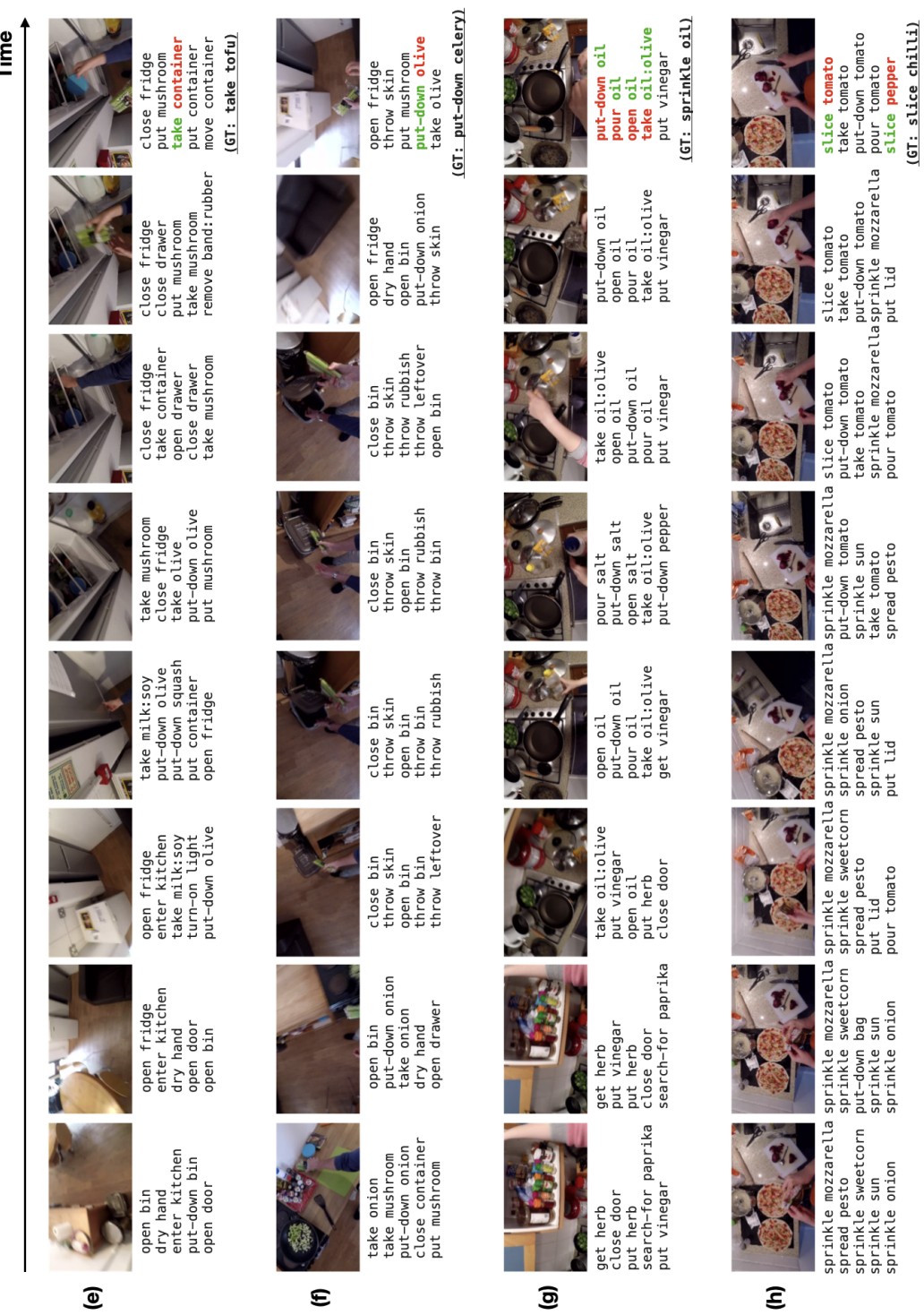

Figure 12: The figure presents four video clips from the **EPIC-Kitchens-55** validation set, showcasing **incorrect** predictions. Each clip depicts the last eight frames and the associated top-5 action anticipations at $\tau_a = 1s$. The ground truth is revealed at the end, with the correct verb and noun of each action highlighted in green.

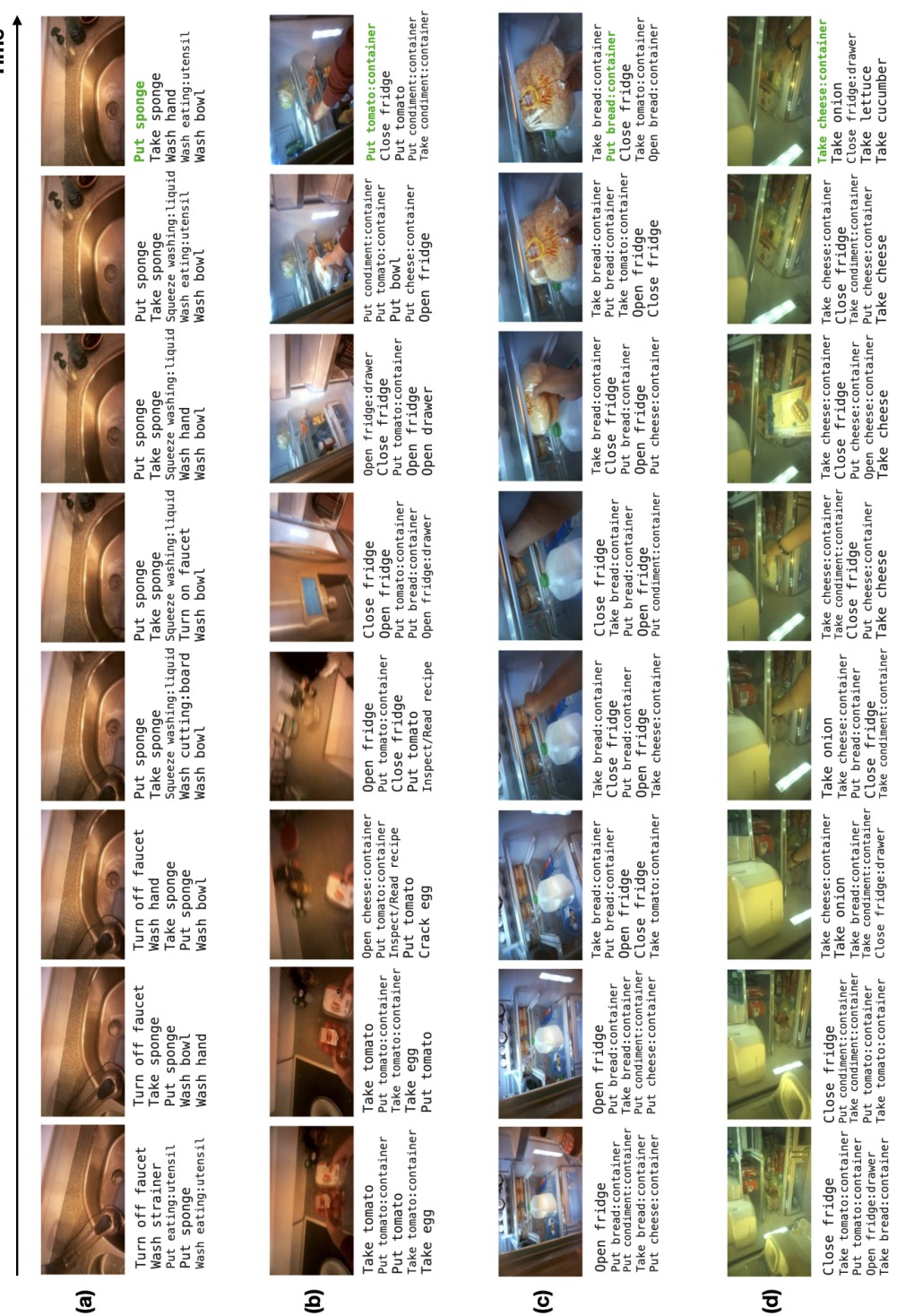

Figure 13: The figure displays four video clips from the **EGTEA Gaze+** validation set, highlighting **correct** predictions. Each clip features the last eight frames along with the corresponding top-5 action anticipations at $\tau_a = 0.5s$. The ground truth is emphasized in bold green.

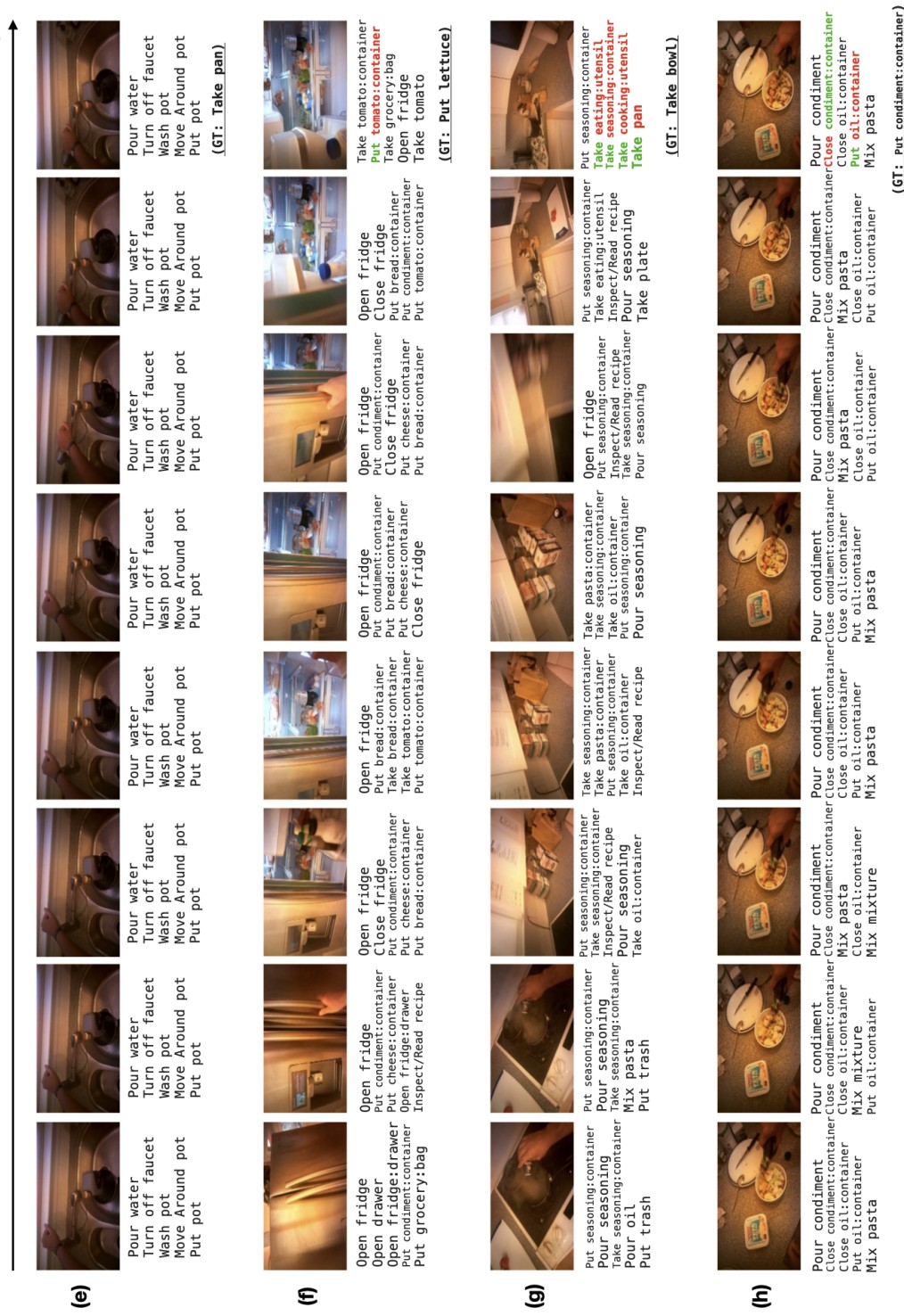

Figure 14: The figure depicts four video clips from the **EGTEA Gaze+** validation set, showcasing **incorrect** predictions. Each clip presents the last eight frames and their corresponding top-5 action anticipations at $\tau_a = 0.5s$. The ground truth is revealed in the final frames, with the correct verb and noun of each action highlighted in green.