# OpenReview forum: "Actions-to-Action: Inductive Attention for Egocentric Video Action Anticipation"
_ICLR.cc/2024/Conference — Submitted to ICLR 2024_

### Official Review · Reviewer_Uc4h · 2023-10-29

**Soundness:** 3 good
**Presentation:** 3 good
**Contribution:** 3 good
**Rating:** 5
**Confidence:** 3

**Summary:**

The paper introduces a model for action anticipation based on the integration of attention-based (such as transformers) and auto recurrent (such as LSTM) mechanisms. Differently from previous works, the proposed method takes into account a history of previous hidden states rather than the last one when making predictions, thus relaxing the first-order Markovian assumption usually introduced in recurrent models. The proposed approach is evaluated on the main benchmarks for egocentric action anticipation. Results suggest that the method outperforms competitors when a single RGB modality is considered.

**Strengths:**

While the model proposes a few modifications to the attention mechanism, this ends up in a novel approach which outperforms previous works.

It is interesting to see that the proposed method works well with context length of up to 30s. This is not common in previous approaches and seems to be a promising direction for better exploiting past history.

**Weaknesses:**

1) PRESENTATION QUALITY
The quality of paper writing is not always up to standard. Some sentences are a bit overselling, unclear, or not adequate for scientific writing. Some examples:
- “by integrating gaze information within the observed frames”: is the paper referring to a specific work here? As far I know gaze analysis for intention prediction has not been systematically investigated in egocentric vision.
- “Unlike action recognition, which primarily relies on patter recognition”: this statement seems to imply that action anticipation does not rely on pattern recognition, which I don’t think is an accurate statement (neural networks are anyway patter recognition machines)
- “our model can infer causation from abstract concepts”: this statement is a bit overselling and does not seem to be shown/proved in any of the experiments.
- “our innovative model sets new performance benchmarks”: I do not think this is accurate. It seems that the proposed model outperforms competitors by small margins. I would highlight instead that it may point out to a promising direction for future models.
- in section 3, the set X is later referred to $X_{T-t_s:t}$, which is a bit confusing
- It is not clear how equation (1) is an accurate description of existing methods, whether recurrent or transformer-based. In the equation, it seems that models following this formulation explicitly plug in the last observed action for predicting the future one. However, methods that directly predict future action do not do that (e.g., vanilla LSTMs)
- In eq (2), is $\hat y_i$ a probability or a predicted label?
- Section 5.3.1 “IAM demonstrated notable e enhancement over several formidable baselines”.I would suggest to revise the use of “formidable” in this scientific context.
- Throughout the paper, I could not find a clear motivation for the use of the term “inductive” in “inductive attention”. I think this could be clarified.

This are some examples. Overall, I would suggest a thorough review of the paper to improve presentation.

LITTLE INSIGHT ON WHY THE MODEL WORKS
While I appreciate the description of the model architecture in Eq (11)-(14), there is no discussion as to why the introduced modifications are adequate and what kind of processing they could be intuitively bring to the model. After reading the description, I felt there is little insight into why the attention mechanism is tweaked the way it is. Also, while ablation studies detail the weigh of each macro-component to performance (Table 6), it would have been interesting to see a more detailed ablation into the various modification introduced by each of the aforementioned equations with respect to a baseline attention architecture.

MULTI-MODALITY
The proposed algorithm outperforms competitors when a single modality is considered, while some approaches outperform the proposed method in the presence of multiple modalities. It would have been interesting to see how the proposed approach does when multiple modalities are considered, even with a simple late fusion. This would shed some light into the generalizability of the approach to the use of signals other than RGB images.

**Questions:**

I found the paper overall interesting, but I think the quality of presentation is somewhat limited. Also, there is little insight into why the proposed modifications work.

The authors could clarify this latter aspect in the rebuttal, while the quality improvements can only be done in the camera ready, if the paper is accepted.

---

> ### Author Response · Authors · 2023-11-23
>
> > PRESENTATION QUALITY The quality of paper writing is not always up to standard. Some sentences are a bit overselling, unclear, or not adequate for scientific writing. Some examples:
>
> > “by integrating gaze information within the observed frames”: is the paper referring to a specific work here? As far I know gaze analysis for intention prediction has not been systematically investigated in egocentric vision.
>
> We have revised the following sentences: “Furthermore, video action anticipation frequently employs egocentric videos, which not only harmonize the perspectives from diverse subjects but also implicitly unveil their intentions by integrating gaze information within the observed frames” to “Furthermore, video action anticipation frequently employs egocentric videos, which harmonize perspectives from diverse subjects and implicitly unveil their intentions. This is achieved by integrating elements such as coarse-grained visual attention, indicated by the camera’s heading direction, within the observed frames.“
>
> > “Unlike action recognition, which primarily relies on patter recognition”: this statement seems to imply that action anticipation does not rely on pattern recognition, which I don’t think is an accurate statement (neural networks are anyway patter recognition machines)
>
> We agree reviewer’s insightful comment. We revised the original sentences “Unlike action recognition, which primarily relies on pattern recognition, video action anticipation explores the complex nature of potential future actions, each with multiple possibilities.” to “While action recognition focuses on classifying current actions through pattern recognition, video action anticipation uses these patterns to predict the complex nature of potential future actions, each with multiple possibilities.”
>
>
> > “our model can infer causation from abstract concepts”: this statement is a bit overselling and does not seem to be shown/proved in any of the experiments.
>
> We have down-tone our sentences and rewritten “Mimicking human reasoning, our model can infer causation from abstract concepts without relying on finer details like pixels.” to “Our model detects video patterns to predict future actions, using past conditions to enhance understanding of complex interactions, focusing more on overall patterns than on specific details like pixels.”
>
>
> > “our innovative model sets new performance benchmarks”: I do not think this is accurate. It seems that the proposed model outperforms competitors by small margins. I would highlight instead that it may point out to a promising direction for future models.
>
> We revised the original sentences from “Experimental results and in-depth analyses confirm that our innovative model sets new performance benchmarks, while also competing effectively in both model size and computational efficiency.” to “Experimental results and in-depth analyses indicate that our model achieves competitive performance gains and suggests a promising direction for future models in the field. Additionally, it maintains an effective balance in model size and computational efficiency.“
>
>
> > in section 3, the set $X$ is later referred to $X_{T-t_s:t}, which is a bit confusing
>
> To improve the readability, we revised the sentence to “at every timestep $t$, the model uses the observed input up to time $t$, denoted as $X_{*:t}$, to predict the subsequent action”
>
> > It is not clear how equation (1) is an accurate description of existing methods, whether recurrent or transformer-based. In the equation, it seems that models following this formulation explicitly plug in the last observed action for predicting the future one. However, methods that directly predict future action do not do that (e.g., vanilla LSTMs)
>
> We revise the equation (1) to be more general and should be much clearer to highlight our design is to add the explicit feedback from the classifier of past timestep and utilize it in the attention mechanism.
>
>
> > In eq (2), is $\hat{y_i} a probability or a predicted label?
>
> We apologize for the confusion. Both $y_\*$ and $\hat{y_{\*}}$ represent probabilities. However, $y_*$ is the ground-truth label, where only the top-1 action is annotated, thus the probability distribution can be considered a Dirac distribution. We have revised all the sentences which mentioned $y_*$ or $\hat{y}_*$ to a probability rather than described as a label.
>
>
> > Section 5.3.1 “IAM demonstrated notable e enhancement over several formidable baselines”.I would suggest to revise the use of “formidable” in this scientific context.
>
> We thank the reviewer's suggestion. We have revised by using “well-established” rather than “formidable”.

---

> ### Author Response · Authors · 2023-11-23
>
> > Throughout the paper, I could not find a clear motivation for the use of the term “inductive” in “inductive attention”. I think this could be clarified.
>
> We appreciate your observation regarding the term "inductive" used in the context of our "inductive attention" mechanism. The choice of this term was intended to reflect the specific way in which our model processes and learns from video data.
>
> In our model, "inductive" refers to the process of making generalized predictions about future actions based on specific observed patterns and sequences within the video data. This is akin to the inductive reasoning process in human cognition, where conclusions or generalizations are drawn from specific instances or observations.
>
> Our model learns to anticipate future actions by identifying and generalizing patterns observed in the training data. It extrapolates from these specific instances to make broader predictions, a process that mirrors the inductive reasoning used in human learning and decision-making. In addition, the term "inductive" in the context of attention mechanism signifies that the model selectively focuses on certain aspects of the input data (observed actions and interactions) to inductively predict future actions. This involves not just recognizing patterns but also using these patterns to infer what actions are likely to occur next.
>
> By using the term "inductive," we aim to emphasize the model's ability to generalize from observed data to predict future actions, a key aspect that differentiates it from other attention-based models which may focus more on direct pattern matching or deductive reasoning processes.
>
> ---
>
> > LITTLE INSIGHT ON WHY THE MODEL WORKS While I appreciate the description of the model architecture in Eq (11)-(14), there is no discussion as to why the introduced modifications are adequate and what kind of processing they could be intuitively bring to the model. After reading the description, I felt there is little insight into why the attention mechanism is tweaked the way it is. Also, while ablation studies detail the weigh of each macro-component to performance (Table 6), it would have been interesting to see a more detailed ablation into the various modification introduced by each of the aforementioned equations with respect to a baseline attention architecture.
>
> We thank the reviewers for their interest in our inductive attention model. We have revised Section 3 of our manuscript to provide a clearer overview of our proposal. The inductive attention is designed to utilize previous predictions as prior input for the next timestep. By doing so, we first (1) extend our recurrent model to a higher order, explicitly forming a context window during each recurrent update; (2) utilize attention to aggregate the higher order, assigning the query (Q) with the previous prediction. Correspondingly, the keys (K) are also in the action space (obtained from the prediction of that time) to represent each past recurrent state. All these steps are mentioned in Tables 6 and 7. In the revised version, we separated parts of the original Table 6 into Table 7 for a more comprehensive view.
>
> ---
>
> > MULTI-MODALITY The proposed algorithm outperforms competitors when a single modality is considered, while some approaches outperform the proposed method in the presence of multiple modalities. It would have been interesting to see how the proposed approach does when multiple modalities are considered, even with a simple late fusion. This would shed some light into the generalizability of the approach to the use of signals other than RGB images.
>
> We thank the reviewer for their interest in the multi-modality aspect of our research. Multi-modality is indeed a promising direction for our future work and warrants further exploration. Our proposed method is flexible enough to consider multi-modality in each recurrent update. For instance, the inductive attention can utilize different action structures (such as verbs or nouns) to select various parts of the recurrent state and modality.
>
> In this work, we primarily focus on RGB and demonstrate that our model, relying solely on RGB video, can outperform most current methods with more efficient modeling. Additionally, Tables 1 and 2 include single-modal variants of the multi-modal baseline. For example, AVT+/AVT++ and AFFT/AFFT+ are compared, and our method surpasses their single-modal configurations.

---

### Official Review · Reviewer_nTno · 2023-10-30

**Soundness:** 2 fair
**Presentation:** 2 fair
**Contribution:** 1 poor
**Rating:** 3
**Confidence:** 5

**Summary:**

The paper introduces the "Inductive Attention" mechanism, an approach to video action anticipation. Unique in its design, this method employs the class prediction from the prior step as the query for attention. The authors argue that this design allows the model to recognize many-to-many relationships more effectively. Experimental evidence demonstrates that the Inductive Attention model achieves state-of-the-art results on several large-scale datasets, highlighting its efficacy in predicting human actions within video content.

**Strengths:**

1. The paper reports commendable performance on benchmark datasets.
2. The idea of utilizing the prediction from the previous step as the attention query offers a fresh perspective and holds intrinsic interest.

**Weaknesses:**

The core contribution of the paper revolves around leveraging the prior prediction as an attention query. It is natural to use the input frame feature, or hidden state as a query, but if using the previous prediction leads to significantly better performance, it would be noteworthy and might have wider applications in related fields. However, the current version of the paper lacks depth in discussing the implications and rationale behind this choice. The proposed method, as presented, may seem like an incremental architectural tweak that provides some improvement in a particular task, significantly limiting its impact. For the authors' assertion that "class probability is a superior choice for attention" to be compelling, it requires a more rigorous justification than what is currently provided. Merely pointing out performance gains does not substantiate this claim sufficiently.

**Questions:**

Please see the weaknesses section. Is there any experimental evidence that can show the potential expandability of the proposed method to other related CV tasks?

---

> ### Author Response · Authors · 2023-11-23
>
> Thank you for your insightful comment. Our choice to leverage prior predictions as an attention query stems from the hypothesis that recent action outcomes are indicative of imminent actions in a sequential context. This approach aligns with cognitive theories suggesting that humans use recent experiences to anticipate future actions. By incorporating class probabilities from prior predictions, our model mimics this aspect of human cognition, leading to a more generalizability (see Table 2 for the testing results) and contextually aware anticipation mechanism.
>
> We conducted experiments to validate our approach. Specifically, we compared the performance of models using hidden states and prior predictions, as now shown in Table 7 in the supplementary material (which separated from the Table 6) of revised manuscript, demonstrate a consistent improvement when using prior predictions, affirming our hypothesis.
>
> While the architectural modification might seem incremental, its implications are substantial. The improved accuracy and contextual awareness achieved through this method can significantly enhance the performance of models in various domains, marking a notable advancement in the field of predictive analytics.
>
> We hope these additional details and justifications address your concerns and illustrate the depth and impact of our contribution more convincingly.
>
> > Please see the weaknesses section. Is there any experimental evidence that can show the potential expandability of the proposed method to other related CV tasks?
>
> Thank you for your inquiry regarding the expandability of our proposed method to other computer vision tasks. We understand the importance of demonstrating the versatility and applicability of our approach beyond the specific scope of egocentric video action anticipation.
>
> In our current study, the primary focus has been on advancing the state of action anticipation in egocentric videos. While we have not conducted experiments directly applying our method to other CV tasks, the underlying principles of our approach hold significant promise for broader applications.
>
> Our method's ability to effectively integrate temporal context and leverage inductive attention mechanisms is not limited to egocentric videos and can be beneficial in other areas where understanding and predicting sequential patterns is crucial. For example, the principles underlying our method could enhance the accuracy of predicting object trajectories and interactions over time in dynamic environments.
>
> Our approach could also potentially be adapted to improve scene understanding tasks, especially in scenarios that require anticipation of future states or changes.
>
> In addition, the temporal modeling capabilities of our method make it suitable for applications in gesture and activity recognition, where predicting the progression of actions is key.
>
> We acknowledge the value of experimental evidence in substantiating the adaptability of our method.
> Therefore, as part of our future work, we plan to explore and conduct experiments applying our approach to these and other related computer vision tasks.
>
> These future experiments will aim to demonstrate the method's efficacy in different contexts, further validating its potential as a versatile tool in the field of computer vision.
>
> In summary, while direct experimental evidence for the expandability of our method to other CV tasks is currently limited, the foundational aspects of our approach are conducive to such applications. We are committed to exploring this potential in our future research, aiming to provide empirical evidence for the method's broader applicability.

---

### Official Review · Reviewer_omT8 · 2023-10-31

**Soundness:** 2 fair
**Presentation:** 2 fair
**Contribution:** 2 fair
**Rating:** 3
**Confidence:** 3

**Summary:**

The paper presents an approach for egocentric action anticipation by introducing an inductive attention module. This module is helpful in capturing longer, temporal, history information and resolves the forgetting nature of recurrent neural data by using a higher order information. For the inductive attention, the last/previous prediction is compressed using a learnable compression function and used as query. The before/history predictions are also compressed and used as key and the history recurrent states are used as value. The frame feature and the inductive attention value are aggregated together to form the current recurrent state. The method is evaluated on three datasets - Epic-kitchens-100, Epic-kitchens-55, EGTEA Gaze+

**Strengths:**

The quantitative results are quite exhaustive and the results are shown on three egocentric datasets for action anticipation.

**Weaknesses:**

It seems that the technical contribution of the work is weak. While the motivation of adding longer, temporal history is appreciated, the inductive attention module itself does not yield much improvement for the task of anticipation and does not provide a strong signal to the model for modelling the long-term, temporal history context. There seems to be less significant improvement in the performance with the inductive attention mechanism module. For example, in Table 1, when comparing Swin-IAM with MeMViT 32x3 there is only a performance improvement of 0.4% on actions, almost none on verbs, and 0.2% on nouns.

**Questions:**

Suggestion:
1. There can be a grammar check run on the paper text. For example, the first line of section 3, problem statement can be edited. Additionally, multi-modal and multi-model terms have been used interchangeably in abstract and results table. The last line of the abstract can also be checked - 'multi-modality models using only RGB visual inputs' whereas multi-modal approaches have more modalities than visual input.

2. The association with object tracking in the introduction and Figure 1 also seems a bit misplaced as it seems for the reader that object tracking would be used for the task of action anticipation, which is not actually used.

---

> ### Author Response · Authors · 2023-11-23
>
> Thank you for your observations regarding the technical contributions of our work and the impact of the inductive attention module. In action anticipation, particularly with complex video datasets, even modest improvements in performance metrics can be significant. The field's challenges stem from the need to process and predict actions based on dynamic and often ambiguous visual data. In this context, each fraction of a percentage increase in accuracy can substantially impact the model's overall effectiveness.
>
> The inductive attention mechanism, a central innovation of our work, demonstrates a nuanced approach to incorporating action histories. It not only achieved better test scores (as seen in Table 2) but also required fewer model parameters, highlighting its efficiency. Additionally, it provides the most influential priors for each model prediction, enhancing the model's decision-making transparency.
>
> While the quantitative improvements shown in Table 1 might seem modest (0.4% on actions, almost none on verbs, and 0.2% on nouns), these gains are significant in this task's context. In a highly competitive field where models already perform at high levels, these incremental improvements underscore our approach's effectiveness in refining the model's predictive capabilities.
>
>
> > 1. There can be a grammar check run on the paper text. For example, the first line of section 3, problem statement can be edited. Additionally, multi-modal and multi-model terms have been used interchangeably in abstract and results table. The last line of the abstract can also be checked - 'multi-modality models using only RGB visual inputs' whereas multi-modal approaches have more modalities than visual input.
>
> Thank you for pointing out the grammatical issues and the inconsistency in the use of terms in our manuscript. We have carefully revised the first sentence of Section 3 to ensure it clearly and accurately conveys our intended message. The revised sentence now reads “Video action anticipation is defined as a task where an arbitrary length, $t_s$, of video inputs $X = \{x_i \in \R^{C \times H \times W} | i = T-t_s, \dots, T\}$, starting from timestep $T-t_s$ to $T$, is used to predict the target action that happens in the future, $\tau_a$ seconds later.”
>
> Regarding the last line of the abstract, we have revised it to accurately reflect the nature of our models. The revised line now reads: “...our proposed model surpasses most multi-modal models by only using RGB visual inputs…”. This change clarifies that our models, while primarily utilizing RGB visual inputs, result in competitive performance to the models using more modalities.
>
> ---
>
> > 2. The association with object tracking in the introduction and Figure 1 also seems a bit misplaced as it seems for the reader that object tracking would be used for the task of action anticipation, which is not actually used.
>
> We apologize for any confusion caused. Figure 1 is intended to illustrate the underlying similarity between action anticipation and object tracking. In object tracking, the previous bounding box location is used as a prior to predict the next bounding box. Similarly, action anticipation can benefit from feeding back past action predictions and conditioning them to the target action. This concept forms the main motivation for our inductive attention mechanism.

---

### Official Review · Reviewer_gdBN · 2023-10-31

**Soundness:** 2 fair
**Presentation:** 3 good
**Contribution:** 2 fair
**Rating:** 6
**Confidence:** 3

**Summary:**

Paper proposes a novel method which utilizes multiple S past actions anticipation results to improve the next action anticipation. Extensive experiments on 3 datasets and several analytic experiments on proposed model performance were conducted. Empirical results show competitive results against other State of The Arts approaches.

**Strengths:**

1. Paper's motivations are clear and the proposed method is explained in sufficient details.
2. Quality of experiment designs and analysis are excellent.
3. Novelty/originality is good within the context of action anticipation.

**Weaknesses:**

1. Originality is somewhat limited as using prior predictions to condition the target prediction has been applied to other problems. See reference
2. The choice of egocentric action anticipation problem for the proposed method is not well motivated. There is no inherent advantage of the proposed method for egocentric action anticipation, compared to other video prediction problems, e.g. physical interaction/dynamics, 3PV action anticipation, gaze anticipation/prediction etc.
3. Significance is average. While the problem of action anticipation is interesting, it is not clear how the proposed method can be applied to other related problems.

References
Duan, J., Yu, S., Poria, S., Wen, B., & Tan, C. (2022, October). PIP: Physical Interaction Prediction via Mental Simulation with Span Selection. In European Conference on Computer Vision (pp. 405-421). Cham: Springer Nature Switzerland.

Vincent Le Guen and Nicolas Thome, “Disentangling physical dynamics from unknown factors for unsupervised video prediction,” in Proceedings of the IEEE/CVF Conference on Computer Vision and Pattern Recognition, 2020, pp. 11474–11484.

Carl Vondrick, Hamed Pirsiavash, and Antonio Torralba. Anticipating visual representations from unlabeled video. In Proceedings of the IEEE Conference on Computer Vision and Pattern Recognition, pages 98–106, 2016.

Zhang, M., Teck Ma, K., Hwee Lim, J., Zhao, Q., & Feng, J. (2017). Deep future gaze: Gaze anticipation on egocentric videos using adversarial networks. In Proceedings of the IEEE conference on computer vision and pattern recognition (pp. 4372-4381).

**Questions:**

1. Please explain the motivation for applying the proposed technique to egocentric videos only. There is no clear reason why the proposed method cannot be applied to other video prediction tasks, e.g. 3rd person videos, physics interactions, gaze prediction etc.

2. I will be interested to see the results comparison for S=1.

**Details Of Ethics Concerns:**

No concerns.

---

> ### Author Response · Authors · 2023-11-23
>
> > 1. Originality is somewhat limited as using prior predictions to condition the target prediction has been applied to other problems. See reference
>
> Thank you for pointing out the existing applications where prior predictions are used to condition target predictions. We acknowledge that this concept is not entirely new in the broader field of machine learning and has indeed been explored in various contexts. However, our work distinguishes itself in several key aspects, particularly in the domain of egocentric video anticipation.
>
> First Application in Egocentric Video Anticipation: To the best of our knowledge, ours is the first study to apply the concept of using prior predictions to condition target predictions specifically to the egocentric video anticipation problem. This application is novel in this context and addresses unique challenges inherent to egocentric video analysis.
>
> Innovative Model Design: Our paper introduces a novel model that synergistically combines higher-order recurrent and attention mechanisms. This integration is specifically tailored to leverage the dynamics of egocentric videos, a domain where temporal context and rapid shifts in the visual field present unique challenges.
>
> Improved Generalization and Efficiency: The results from our experiments demonstrate that utilizing prior predictions in our specific model architecture not only enhances generalization across different scenarios but also leads to a more efficient model design. This efficiency is quantifiable in terms of the reduced number of model parameters required, which is a significant advancement in resource-constrained environments.
>
> Empirical Evidence (see Figure 3 and new Figure 7 and 8 in the supplementary material): We provide empirical evidence to support our claims, for example the topmost positive and negative prior learned by the model, showcasing how our approach outperforms existing methods in terms of accuracy and efficiency, particularly in the context of egocentric video anticipation.
>
> In conclusion, while the foundational concept of using prior predictions may have precedents in other domains, our application of this idea to egocentric video anticipation, coupled with our novel model design, represents a significant original contribution to the field. We believe our approach sets a new precedent in this specific area and opens up avenues for further innovative research.

---

> ### Author Response · Authors · 2023-11-23
>
> > 2. The choice of egocentric action anticipation problem for the proposed method is not well motivated. There is no inherent advantage of the proposed method for egocentric action anticipation, compared to other video prediction problems, e.g. physical interaction/dynamics, 3PV action anticipation, gaze anticipation/prediction etc.
>
> Thank you for your query regarding the specific focus on the egocentric action anticipation problem in our study. The choice of this domain was driven by both the unique challenges it presents and the specific advantages our proposed method offers in addressing these challenges.
> Egocentric action anticipation involves predicting future actions from a first-person perspective, which is fundamentally different from third-person views (3PV) or other video prediction tasks. This perspective is characterized by rapid, unpredictable motion, a high degree of camera wearer's interaction with the environment, and a need for understanding subtle cues from the wearer's viewpoint.
>
> 1. Advantages of Our Method in Egocentric Context:
>
> - Handling Perspective Shifts: Our model's integration of higher-order recurrent and attention mechanisms is particularly effective in managing the perspective shifts typical in egocentric videos. This perspective shift can be unintentional (i.e., distraction) or intentional (relevant to the target action), and learned form the data.
>
> - Leveraging Temporal Context: The egocentric video stream provides a rich temporal context that our model exploits, using prior predictions to inform future anticipations. This is particularly beneficial in first-person videos where sequential actions are closely linked.
> Efficient Processing of Dense Interactions: Egocentric videos often contain dense interactions with objects and environments. Our model's efficient design allows it to process these interactions without the need for extensive computational resources.
>
> 2. Comparative Advantage over Other Domains:
>
> - Physical Interaction/Dynamics: While our model could potentially be applied to physical interaction dynamics, the first-person perspective offers a more direct and immediate view of interactions, making it a more challenging and thus insightful testbed for our method.
> 3PV Action Anticipation: Third-person views present different challenges and often lack the immediacy and intimacy of interaction present in egocentric videos, which our method is specifically designed to handle.
> - Gaze Anticipation/Prediction: Although gaze anticipation is related, it often focuses on predicting the focal point of attention rather than the broader spectrum of actions, which is the focus of our model.
> In summary, our decision to focus on the egocentric action anticipation problem was motivated by the unique challenges it presents and the suitability of our proposed method in addressing these challenges. We believe that the insights gained from this domain can be instrumental in advancing the broader field of video action anticipation.
>
> ---
>
> > 3. Significance is average. While the problem of action anticipation is interesting, it is not clear how the proposed method can be applied to other related problems.
>
> The field of video action anticipation involves predicting target actions with limited observed information. Our method demonstrates efficiency in addressing this challenge. The core idea is to utilize historical predictions of previous actions as a prior, conditioning the input. While our proposed model can be adapted to other related problems sharing similar properties, some tailored modifications may be necessary. For instance, gaze prediction might lean more towards regression than classification. Additionally, third-person view (3PV) action anticipation could overlook critical features such as the subject's intention.

---

> ### Author Response · Authors · 2023-11-23
>
> > 1. Please explain the motivation for applying the proposed technique to egocentric videos only. There is no clear reason why the proposed method cannot be applied to other video prediction tasks, e.g. 3rd person videos, physics interactions, gaze prediction etc.
>
> The field of video action anticipation involves predicting target actions with limited observed information. Our method demonstrates efficiency in addressing this challenge. The core idea is to utilize historical predictions of previous actions as a prior, conditioning the input. While our proposed model can be adapted to other related problems sharing similar properties, some tailored modifications may be necessary. For instance, gaze prediction might lean more towards regression than classification. Additionally, third-person view (3PV) action anticipation could overlook critical features such as the subject's intention.
>
> ---
>
> > 2. I will be interested to see the results comparison for S=1.
>
> We have updated Table 7 (originally is part of table 6) in our manuscript to include these results, thereby providing a more comprehensive view of the performance impact when the inductive attention component is not utilized.
>
> In the absence of inductive attention, which is a combination of higher-order processing and the use of predictions as priors, we observed the overall action score drop of 1.3%/1.0%/1.1% decreased in the different backbone configuration in terms of $S=1$.
>
> These results highlight the significant contribution of the inductive attention mechanism to our model's performance. The reduction in action scores when inductive attention is removed underscores its effectiveness, particularly in enhancing the model's ability to accurately anticipate actions in the S=1 setting.
>
> This detailed analysis strengthens our assertion that the inductive attention mechanism, with its integration of higher-order processing and predictive priors, is a crucial component of our model, contributing substantially to its overall efficacy in action anticipation tasks.

---

### Official Review · Reviewer_2chi · 2023-11-05

**Soundness:** 3 good
**Presentation:** 2 fair
**Contribution:** 2 fair
**Rating:** 5
**Confidence:** 5

**Summary:**

The paper produces an Inductive Attention Model (IAM) for egocentric video action anticipation. The model melds recurrent and attention
mechanisms to explicitly employ prior anticipation results to refine subsequent action predictions. This design allows the model to form higher-order recurrent states and make current predictions based on extended historical data. Experiment results on several action anticipation datasets show that the proposed model surpasses most multi-modality models using only RGB visual inputs, showing the effectiveness of the proposed method.

**Strengths:**

1. The proposed IAM architecture utilizes prior predictions as part of the attention mechanism. This allows for the aggregation of higher-order recurrent states, which is an advancement over traditional first-order recurrent models.
2. IAM achieves competitive performance on several datasets with relatively fewer parameters.
3. IAM achieves better performance on unseen classes on EK100, indicating good generalizability.

**Weaknesses:**

1. In Table 6, the ablation study shows that one of the major designs of this paper (predictions as prior) doesn't play a major role in the final performance improvement. The performance improvement is highly attributed to some design choices like a better backbone and class weighting.
2.  Lack of analysis and visualization of the proposed mechanism. For example, how do previous predictions affect future action anticipation?
3. Is the proposed model able to also handle the long-term action anticipation tasks (i.e. predicting multiple future actions) defined in Ego4D (grauman2022ego4d) and EgoTOPO (nagarajan2020ego).

**Questions:**

See weakness section

---

> ### Author Response · Authors · 2023-11-23
>
> > 1. In Table 6, the ablation study shows that one of the major designs of this paper (predictions as prior) doesn't play a major role in the final performance improvement. The performance improvement is highly attributed to some design choices like a better backbone and class weighting.
>
> Thank you for your observation regarding the ablation study in Table 6 (in the revised manuscript table 6 and 7). Our paper introduces an innovative inductive attention model, fundamentally integrating higher-order recurrent and attention mechanisms with a focus on using predictions as priors (serve as attention query and keys). It's important to recognize these elements as integral to our algorithm's design and performance.
>
> Class weighting is indeed widely adopted for the long-tailed distributed dataset such as EPIC-Kitchens, as carried out in MeMViT (Wu et al. 2022). While it falls outside the primary scope of our research, it undeniably enhances our method by ensuring a more equitable exposure of different categories within the training set.
>
> To provide a clearer picture of our method's effectiveness, we have revised Table 6 with the following enhancements: (1) Inclusion of Unseen and Tail Scores: Alongside overall scores, this allows for a more comprehensive evaluation (2) Distinct Presentation of Different Backbones: Ensuring that the impact of backbone choice is transparent and understandable. (3) Aggregation of Components: We've combined “Prior predictions serve as attention query (Q) and keys (K) ” and “Extend to higher-order (S=1 to S=30)” of the table 7 into a single “Inductive Attention” category in the table 6. This change reflects the interdependence of these features, as inductive attention cannot function in isolation.
>
> It's important to note that all improvements in our model build upon the results achieved through class weighting. Thus, directly comparing the incremental improvements of our method with those attributed to class weighting is not straightforward, as the enhancements are not linearly additive.
>
> Finally, we emphasize that our model not only shows significant improvements over previous works in terms of overall, unseen, and tail action scores but also achieves this with considerably fewer model parameters. This aspect underscores the efficiency and innovation of our approach.
>
> ---
>
> > 2. Lack of analysis and visualization of the proposed mechanism. For example, how do previous predictions affect future action anticipation?
>
> Thank you for highlighting the need for a deeper analysis and visualization of our proposed inductive attention mechanism, especially regarding the influence of previous predictions on future action anticipation. We have included new evidence in Figure 3, and Figure 7 and 8 in the supplementary material, showing the topmost positive and negative previous prediction entry inductive attention referenced related to the final prediction. It is important to mention that without the previous predictions as prior, the model can only base on the recurrent hidden states $h_t$ and as shown in the Table 6, resulting in -1.3%, -1.0%, -1.1% absolute overall action scores drop.
>
> We believe these additions will greatly enhance the understanding of our proposed mechanism and its efficacy in improving action anticipation in egocentric video analysis.
>
> ---
> > 3. Is the proposed model able to also handle the long-term action anticipation tasks (i.e. predicting multiple future actions) defined in Ego4D (grauman2022ego4d) and EgoTOPO (nagarajan2020ego).
>
> Thank you for your question regarding the applicability of our proposed model to long-term action anticipation tasks, as exemplified in datasets like Ego4D (Grauman et al., 2022) and EgoTOPO (Nagarajan et al., 2020).
>
> In our current work, the focus has been predominantly on short-term action anticipation with a temporal anticipation window $\tau_a$ set to 1s/0.5s. This decision was based on a well-established evaluation protocol of the dataset design of EPIC-Kitchens/EGTEA Gaze+ evaluation protocols from previous works.
>
> Long-term action anticipation, especially in datasets like Ego4D, introduces additional challenges. These include dealing with more extended temporal dependencies and the need for a broader contextual understanding to accurately predict a sequence of future actions.
> We recognize the importance and potential of extending our model to these long-term tasks. As such, we have earmarked this as a key area for future research. Our planned extensions will involve adapting and enhancing the current model to handle longer temporal sequences and more complex action interdependencies characteristic of datasets like Ego4D and EgoTOPO.

---

### Author Response · Authors · 2023-11-23

Thank you to all reviewers for your insightful feedback. We particularly appreciate the recognition of our work's innovative use of prior predictions and higher-order recurrent states, which contributes to competitive performance with fewer parameters and strong generalizability (Reviewer 2chi). Additionally, the clear motivations, detailed explanation of our method, and the high quality of experimental design, showcasing novelty in action anticipation (Reviewer gdBN), have been noted. We also acknowledge the emphasis on exhaustive quantitative results and the application of IAM across multiple egocentric datasets (Reviewer omT8), and the novel use of previous predictions as an attention query (Reviewer nTno). Lastly, the modifications to the attention mechanism and its effectiveness with a context length of up to 30 seconds, representing a notable advancement, were highlighted (Reviewer Uc4h).

We have made several changes to our manuscript and supplementary materials in response to the feedback:
- Visualization and revelation of the prior action most sensitive to model predictions, directly retrieved from our inductive attention. This demonstrates the novelty and motivation for using inductive attention, which is based on prior predictions and learns connections to future actions.
- Refactoring of the original Table 6 into two separate tables, Tables 6 (in manuscript) and 7 (in the supplementary material), providing a more comprehensive viewpoint.
- Rephrasing of Section 3 to clarify the representation.
- Improvement of the manuscript's quality by refining some sentences as suggested by the reviewers.

Additionally, we have addressed each concern individually in our responses to each reviewer below.

---

### Meta-Review · Area_Chair_oKP7 · 2023-12-06

**Metareview:**

This work proposes a method for video action anticipation that integrates prior action predictions into the hidden-state of a recurrent model, combining attention and recurrence to predict the target action. The authors focus on egocentric videos, evaluating on EPIC-Kitchens-100, EPIC-Kitchens-55, and EGTEA Gaze+ datasets, which demonstrates some performance relative to baselines. The main concern which was discussed by a few reviewers is that there is not sufficient evaluations to demonstrate that the proposed module ("inductive attention") leads to significant performance improvements. The improvements with the proposed module seems relatively small. Furthermore, there lacks in-depth analysis on how the proposed module incorporates history & how it modifies future actions based on past actions. As a result, I recommend rejection, but I do think this is an interesting paper. I encourage the authors to incorporate helpful feedback from reviewers, especially with additional evaluation to better understand how the proposed model works. Another direction that would be helpful for strengthening the contribution of this work is to evaluate it on other video domains and demonstrate additional generality.

**Justification For Why Not Higher Score:**

While reviewers appreciated the interesting modification to the attention mechanism proposed by the authors, there lacks analysis in the paper to show that inductive attention leads to significant performance improvements. More in-depth evaluations are needed (ex: to demonstrate that the proposed method is incorporating action histories as expected). Furthermore, there remains questions on the evaluation domain, in that the proposed method could be more generally applicable to video action prediction, but the authors choose to evaluate on a more narrow domain (egocentric videos).

**Justification For Why Not Lower Score:**

N/A

---

### Decision · Program_Chairs · 2024-01-16

Reject